# Effects of strengthening the Baltic Sea ECA regulations

Jan Eiof Jonson[1], Michael Gauss[1], Jukka-Pekka Jalkanen[2], and Lasse Johansson[2]

[1]Norwegian Meteorological Institute, Oslo, Norway
[2]Finnish Meteorological Institute, Helsinki, Finland

**Correspondence:** Jan Eiof Jonson (j.e.jonson@met.no)

**Abstract.**

Emissions of most land based air pollutants in western Europe have decreased in the last decades. Over the same period emissions from shipping have also decreased, but with large differences depending on species and sea area. At sea, sulphur emissions in the SECAs (Sulphur Emission Control Areas) have decreased following the implementation of a 0.1% limit on
sulphur in marine fuels from 2015. In Europe the North Sea and the Baltic Sea are designated as SECAs by the International maritime Organisation (IMO).

Model calculations assuming present (2016) and future (2030) emissions have been made with the regional scale EMEP model covering Europe and the sea areas surrounding Europe including the North Atlantic east of 30 degrees west. The main focus in this paper is on the effects of ship emissions from the Baltic Sea. To reduce the influence of meteorological variability,
all model calculations are presented as averages for 3 meteorological years (2014, 2015, 2016). For the Baltic Sea, model calculations have also been made with higher sulphur emissions representative of year 2014 emissions.

From Baltic Sea shipping the largest effects are calculated for $NO_2$ in air, accounting for more than 50% of in central parts of the Baltic Sea. In coastal zones contributions to $NO_2$ and also nitrogen depositions can be of the order of 20% in some regions. Smaller effects, up to 5 – 10%, are also seen for $PM_{2.5}$ in coastal zones close to the main shipping lanes. Country
averaged contributions from ships are small for large countries that extend far inland like Germany and Poland, and larger for smaller countries like Denmark and the Baltic states Estonia, Latvia and Lithuania, where ship emissions are among the largest contributors to concentrations and depositions of anthropogenic origin. Following the implementations of stricter SECA regulations, sulphur emissions from Baltic Sea shipping now have virtually no effects on $PM_{2.5}$ concentrations and sulphur depositions in the Baltic Sea region.

Adding to the expected reductions of air pollutants and depositions following the projected reductions in European emissions, we expect that the contributions from Baltic Sea shipping to $NO_2$ and $PM_{2.5}$ concentrations, and to depositions of nitrogen, will be reduced by 40 – 50% from 2016 to 2030 mainly as a result of the Baltic Sea being defined as a Nitrogen Emission Control Area from 2021. In most parts of the Baltic Sea region ozone levels are expected to decrease from 2016 to 2030. For the Baltic Sea shipping, titration, mainly in winter, and production, mainly in summer, partially compensate. As a result the
effects of Baltic Sea shipping on ozone is similar in 2016 and 2030.

# 1 Introduction

Even though emissions of most air pollutants have decreased in the countries surrounding the Baltic Sea (BAS) in past decades (Tista et al., 2018), air pollution and atmospheric depositions affecting ecosystems remain a problem in the region. Significant contributions to the emissions also come from shipping, both inside and outside the region. Obtaining reliable data on emissions from international shipping has always been challenging, but in recent years ship emissions estimated based on AIS (Automatic Identification System) positioning data have become available, continuously tracking the position of the vessels, resulting in substantial improvements in the reliability of ship emissions data.

A number of IMO (International Maritime Organisation) and EU regulations have been implemented in the recent past, or will be implemented in the near future, affecting ship emissions in European waters. Most noteworthy are the SECA (Sulphur Emission Control Area) regulations, reducing the maximum sulphur content allowed in marine fuels from 1.0 to 0.1% from 1. January 2015 (IMO, 2008). Fuels with higher sulphur content may be used in combination with emission reduction technology reducing sulphur emission to levels equivalent to the use of compliant low-sulphur fuels. In European waters the North Sea (NOS) and BAS are designated as SECAs by the IMO. These two sea areas are also accepted as NECAs ($NO_x$ Emission Control Areas) from 2021 (IMO, 2017). Reductions of $NO_x$ emissions are expected to occur only gradually in the NECAs as these regulations only apply to new ships or when major modifications are made on existing ships. Furthermore, from 2020 a global cap on sulphur content in marine fuels of 0.5% will be implemented.

The global effects of international shipping on air pollution and depositions have been discussed in several papers (Corbett et al., 2007; Endresen et al., 2003; Eyring et al., 2007; Sofiev et al., 2018). In a global model calculation Jonson et al. (2018) found that a large portion of the anthropogenic contributions to air pollution and nitrogen depositions in adjacent countries could be attributed to NOS and BAS ship emissions of $NO_x$ and particles also after the introduction of stricter SECA regulations in 2015. In addition, several regional studies focusing on the effects of NOS and BAS ship emissions have been performed. Jonson et al. (2015) studied the effects of reducing the sulphur content in marine fuels from 1.5 to 1% in 2011 on air pollution, including also calculations of health effects as well as effects of future (2030) ship emissions. They found that the introduction of a NECA from 2016 (later postponed to 2021) would reduce the burden on health due to shipping in the BAS region. Reductions in future $PM_{2.5}$ (particulate matter with diameter less than $2.5\mu$m) levels as a result of the 2021 NECA are also predicted by Karl et al. (2018). Brandt et al. (2013) calculated the effects of ship emission on Europe for the years 2000 and 2020. They found that the implementation of the stricter SECA regulations in the BAS and the NOS would result in substantial health improvements in Europe. Karl et al. (2019) compared the effects of BAS shipping calculated by three different chemistry transport models using year 2012 emissions and meteorology. They found that in the entire BAS region the average contribution from ships to $PM_{2.5}$ is in the range of 4.3 - 6.5% for the three CTMs, and deposition of oxidised nitrogen to the Baltic Sea in the 20 - 24ktN per year range. Claremar et al. (2017) calculated the dispersion of air pollutants and depositions from NOS and BAS shipping for the period 2011 to 2050 with the main focus on sea-water acidity in BAS. They found that, also in the future, ship emissions could remain a major source of acidity, in particular when assuming high penetration of open loop scrubbers in combination with the use of high sulphur-content fuels.

SO$_x$ removal by scrubbing the exhaust can significantly reduce both the gaseous sulphur compounds as well as particulate matter. Scrubbers may use seawater as a cleaning agent if the alkalinity of seawater is high enough and contains enough carbonates, bicarbonates and borates. However, in areas of low alkalinity, like the Bothnian Bay in the Baltic Sea, the required wash water volume becomes very large and chemicals, like caustic soda, are added to neutralize the acidic releases. The wash water may also contain other pollutants as heavy metals.

Ship owners can also comply with stringent sulphur rules by using LNG (Liquefied Natural Gas). However, during 2016 only about 0.8% of the energy need of the Baltic Sea fleet was produced with LNG. Use of renewable liquid fuels is rather limited because of high price and low availability. Liquid biofuels are not used by any ship in our modelling approach.

In this paper we have calculated the effects of ship emissions in the BAS on air pollution and depositions of oxidised sulphur and nitrogen in adjacent countries. Calculations have been made applying BAS emissions prior to (2014) and after (2016) the implementation of the stricter SECA regulations, which went into force on 1 January 2015. Furthermore, model calculations have been made with future (2030) land-based and ship emissions.

## 2 Experimental setup

### 2.1 Emissions

Land-based emissions have been provided by the International Institute for Applied Systems Analysis (IIASA) within the European FP7 project ECLIPSE. In this study we use version 5a (hereafter 'ECLIPSEv5a'), a global emission data set on 0.5 x 0.5 degree resolution, which has been widely used in recent years by the scientific community (http://www.iiasa.ac.at/web/home/research/researchPrograms/air/ECLIPSEv5.html, Last accessed: 27 February 2019). ECLIPSEv5a is available in 5-year intervals from 2005 onwards, and in this study we have chosen data for 2015 and 2030.

The ECLIPSE v5a emissions were re-gridded using the TNO-MACC-III 0.125x0.0625 lon-lat emission distribution (Kuenen et al., 2014) for year 2011. During the re-gridding process only the spatial distribution of the ECLIPSE v5a emissions was modified, while the national and sector totals remained unchanged. Where TNO-MACC-III emissions are not available (such as North-Africa) the gridded ECLIPSE v5a emissions were interpolated to the TNO-MACC grid resolution. Any missing sectors for countries which were included in the TNO-MACC-III emission data were also completed from the interpolated ECLIPSE v5a emissions.

In reality land based emissions will change between years. Annual emissions from year 2000 to 2016 for the European countries are listed in EMEP Status Report 1/2018 (2018). In the Baltic region reported changes in country emission are small with the exception of SO$_x$ emissions in Poland dropping almost 20% from 2014 to 2016.

In regard to ship emissions in the BAS, we use emission data as provided by FMI (Finish Meteorological Institute) for the year 2014 (i.e. with 1% maximum sulphur content in fuels in the SECA) and 2016 (maximum sulphur content reduced to 0.1% in the SECA). For the remaining sea areas, ship emissions for year 2015 are used from a previous global data set (Johansson et al., 2017).

The emissions from shipping have been calculated with the Ship Traffic Emission Assessment Model (STEAM) based on ship movements from the automatic identification system (AIS) which provides real time information on ship positions. The model requires as input detailed technical specifications of all onboard fuel-consuming systems and other relevant technical details for all ships considered. The data from IHS Global (2017) constituted the most significant source for this information. The STEAM model is described in Jalkanen et al. (2009, 2012, 2016) and Johansson et al. (2013, 2017). Hourly emission grids for Baltic Sea ship emissions were produced based on vessel-specific modelling, considering the changes in fuel sulphur content that occurred between 2014 and 2016.

In STEAM scrubbers can operate in closed or open loop mode, depending on the equipment installed. If a hybrid scrubber system is known to be installed, it is assumed to operate in open loop mode when the vessel operates in an area where open loop systems are feasible. Closed loop mode of a hybrid scrubber is assumed in the Bothnian Bay and restricted zones, like German waters. If a vessel has open loop scrubber installed and it enters a restriction zone, the model assumes a fuel switch to low sulphur fuels. Emission modelling uses scrubber equipment type (closed/open/hybrid), vessel identity and installation date as input to emission modelling. All future scrubber scenarios introduce hybrid scrubbers to the fleet.

Globally, during 2014 there were 77 vessels using a scrubber, of which 30% were of open loop, 48% of closed loop and 22% of hybrid type. By 2016 scrubber installations were doubled globally to 155 units. In the Baltic Sea area during 2016, there were 85 vessels operating a scrubber releasing 73 million tonnes of wash water to the sea. Almost all of this (99.8%) discharge came from open loop operation of scrubbers.

Ship emitted pollutants were modelled using AIS data for year 2014 and 2016. Any changes in vessel activity, fleet size and development will have an impact on energy use and all pollutant emissions. However, the sulphur rule was the only significant change which had a large impact on emitted pollutants. Both PM and $SO_x$ were reduced by this change, but only the sulphate fraction of PM was reduced accordingly whereas other components of PM were less affected.

From 2021 onward, $NO_x$ emissions for new ships have to comply with IMO Tier 3 regulations. These contributions were taken into account in the emission modelling. Future emission projections for the year 2030 also include changes in:

- energy efficiency improvements, modelled following the method of Kalli et al. (2013), which goes beyond the Energy Efficiency Defined Index (EEDI) requirements of the IMO;

- fleet size increase.

- vessel size growth, assuming a linear annual growth dependent on ship types;

Annual growth rates in fleet size are implemented as percentage increase per type of ship: For example, if the annual percentage growth is n% for container ships we duplicate n% of the container ships in the current fleet in the following year. The following growth rates are assumed for vessel DWT: Vehicle carriers and RoRo: 1.25% per annum; Dry cargo: 0.4% per annum; Container carriers: 1.2% per annum; Liquid cargo: 2.0% per annum; Passenger vessels, ferries, High-speed craft: 0.3% per annum; Cruise ships: 0.3% per annum; Fishing vessels: 0.3% per annum. Vessel size growth for other types were set to zero. For those vessels, the vessels size remains at 2014 level.

As the ship emission data are used for multiple meteorological years (see next section), we did not retain the high (hourly) temporal resolution in the data but rather aggregated them to monthly resolution before use in the chemistry transport model.

## 2.2 Model calculations of air pollutants and depositions

Concentrations of air pollutants and depositions of sulphur and nitrogen have been calculated with the EMEP MSC-W model (hereafter 'EMEP model'), version rv4.14, on 0.1 x 0.1 degrees resolution for the domain between 30 degrees W and 45 degrees E and between 30 and 75 degrees N. A detailed description of the EMEP model can be found in Simpson et al. (2012) with later model updates being described in Simpson et al. (2018) and references therein. The EMEP model is available as Open Source (see https://github.com/metno/emep-ctm, Last accessed: 27 February 2019), and is regularly evaluated against measurements as part of the EMEP status reports. See Gauss et al. (2016, 2017, 2018) for evaluations of the meteorological years 2014, 2015 and 2016, respectively. In addition, the EMEP model has successfully participated in model inter-comparisons and model evaluations presented in a number of peer-reviewed publications Colette et al. (2011, 2012); Angelbratt et al. (2011); Dore et al. (2015); Karl et al. (2019); Stjern et al. (2016); Jonson et al. (2018). Vivanco et al. (2018) evaluated depositions of sulphur and nitrogen species in Europe calculated by 14 regional models, showing good results for the EMEP model.

In the present study the model is driven by meteorological data from the European Centre for Medium-Range Weather Forecasts (ECMWF) based on the CY40R1 version of their IFS (Integrated Forecast System) model. All simulations for this paper have been run for the three meteorological years 2014, 2015 and 2016, and then averaged, in order to cancel out meteorological variability. The simulations are:

- Present_Base: Base case with ship emissions of 2016. Land-based emissions for 2015 (from ECLIPSEv5);

- Present_NoShip: As Present_Base, but without ship emissions in the BAS;

- Present_HiSulphur: As Present_Base, but with ship emissions of 2014 (i.e high sulphur content) in the BAS;

- Future_Base: Ship emissions of 2030 (assuming NECA and business as usual development) and land-based emissions of 2030 (from ECLIPSEv5);

- Future_NoShip: As Future_Base, but without ship emissions in the BAS.

The emissions are also summarised in Table 1. In the future scenarios it is assumed that ships that are in compliance with the NECA regulations will operate the equipment (i.e. be compliant) also when sailing outside the NECA.

## 3 Model results

In this section model results for parts of Europe centred around the BAS are shown. Concentrations and depositions are shown as averages for three meteorological years for Present_Base and Future_Base and for differences between the two Base runs and the perturbation scenarios as described in Section 2.2. The impact on $PM_{2.5}$ levels and on the depositions of oxidised nitrogen

and sulphur species derived from the perturbation model runs presented here, forms the basis for coming papers discussing the effects on human health (Barregård et al., 2019) and assessing the environmental impacts, including the exceedances of critical loads from ship emissions in the BAS (Repka et al., 2019).

In Gauss et al. (2018) the EMEP model results for 2016 compared to measurements are discussed in detail. Although the model setup is not completely identical the results are qualitatively very similar. The model underestimates $NO_2$. Measured $PM_{2.5}$ is also underestimated, and results for the individual $PM_{2.5}$ components are mixed, with $SO_4$ underestimated, whereas other components are overestimated compared to measurements.

## 3.1  Air pollution due to Baltic Sea shipping

Concentrations of $NO_2$ for Present_Base are shown in Figure 1a. The lifetime of $NO_2$ is relatively short, and as a result the concentrations largely reflect the locations of the main source areas. Concentrations of $NO_2$ are high in Central Europe and in and around the English Channel with markedly lower concentrations north and east of the BAS. In the NOS and the BAS the major ship tracks are clearly visible. Figure 1c shows the difference between the Present_Base and the Present_NoShip scenarios The calculations show that ship emissions account for more than 50% of $NO_2$ in central parts of the BAS and for a substantial percentage also in coastal zones, in particular in Denmark, southern parts of Sweden and Finland and the Baltic states (Estonia, Latvia and Lithuania). This is also illustrated in Table 2 where measured $NO_2$ at sites located in the BAS coastal regions are compared to the Present_Base, Present_NoShip and Present_HiSulphur model calculations calculated with 2016 meteorology. The position of the measurement sites and the corresponding time series plots for $NO_2$ are shown in Appendix A. In the Present_NoShip case $NO_2$ levels are clearly underestimated and correlations and RMS errors deteriorated compared to the Present_Base calculation, in particular for those sites located very close to major shipping routes. The comparisons with measurements convincingly show that these measurements can only be reproduced when BAS ship emissions are included. The contributions to individual countries will be further discussed in a later section.

As shown in Table 2, measured $SO_2$ levels for 2016 are relatively well reproduced by the model for the Present_Base calculation. The position of the measurement sites and the corresponding time series plots for $SO_2$ are shown in Appendix A. The effects of excluding the BAS ship emissions in the Present_NoShip scenario have only minor effects on the $SO_2$ levels. Replacing 2016 BAS emissions with 2014 (Present_HiSulphur) has much larger effects, resulting in an overestimation of $SO_2$ levels at most of the sites listed in Table 2, and in particular so for Anholt and Råö, located very close to the shipping routes. This clearly illustrates the effects of the stricter SECA regulations. With the high ship emissions of 2014, the measurements for 2016 can not be reproduced. This is also a strong indication that the ships are largely in compliance with the SECA regulations. As for $NO_2$, the contributions to individual countries are discussed further in a later section

$PM_{2.5}$ in the atmosphere is a mixture of many chemical species of both natural and anthropogenic origins. It is emitted both as a primary pollutant and formed as a secondary pollutant in the atmosphere. As a result $PM_{2.5}$ concentrations are more spread out compared to $NO_2$. Concentrations decrease from south to north from a maximum in central Europe. As shown in Figure 1d the percentage contributions from BAS shipping, calculated as Present_Base – Present_NoShip, are much smaller for $PM_{2.5}$ than for $NO_2$ but with noticeable contributions in coastal zones, in particular in parts of Denmark, Sweden and Finland.

Figure 1e shows higher contributions when assuming BAS shipping at 2014 levels (Present_HiSulphur), prior to the implementation of the stricter SECA regulations. These results are also illustrated in the comparisons of model scenario calculations at the measurement sites located in BAS coastal regions as listed in Table 2. The position of the measurement sites and the corresponding time series plots for $PM_{2.5}$ are shown in Appendix A. For $PM_{2.5}$ differences between the Present_Base and the Present_NoShip cases are much smaller than for $NO_2$. Likewise, differences are smaller than for $SO_2$ between Present_Base and Present_HiSulphur. In Table 2 we also show measured and model calculated concentrations of $SO_4$.

Continuing a downward trend from the late 1980s, land-based sulphur emissions have decreased by more than 50%, i.e. more than for any other of the major air pollutants (Tista et al., 2018) and thus the importance of sulphur in particle formation has thus decreased relative to other anthropgenic emitted species and natural sources. In the SECAs the sulphur content in marine fuels has decreased from the global average of about 2.5% to 1% in 2011 and finally to 0.1% in 2015. As a result of these large emission reductions the fraction of $SO_4$ in $PM_{2.5}$ in the BAS region has decreased even further here. At the sites in Table 2 both the measured and model calculated fraction of $SO_4$ in $PM_{2.5}$ is about 0.15. As $SO_4$ make up a moderate portion of the $PM_{2.5}$ composition this fraction increase only by a small amount with the Present_HiSulphur scenario.

The model results underestimate the measurements at most of the sites listed. Based only on the comparisons between measurements and the different model scenarios for $PM_{2.5}$ one can not conclude that the Present_Base scenario is more realistic than the other two. As for $NO_2$ and $SO_2$, the contributions to individual countries are discussed further in a later section.

## 3.2 Depositions of sulphur and nitrogen from Baltic Sea shipping

Total depositions (wet and dry) of oxidised sulphur and nitrogen for Present_Base are shown in Figure 2a,b. The highest depositions of both sulphur and nitrogen are seen over Central Europe. For nitrogen, high levels of depositions also extend into northern Germany and Denmark. Based on the difference between Present_Base and Present_NoShip a significant amount of the nitrogen depositions can be attributed to BAS shipping (Figure 2c), contributing to more than 15% of the total nitrogen depositions in major parts of the BAS and also in parts of Sweden, Finland and the Baltic states (Estonia, Latvia and Lithuania). Dry deposition is parameterised as a function of sub grid-scale ecosystems and is typically higher than the grid average for forest ecosystems (both coniferous and deciduous). This will affect the calculations of critical loads for acidification and eutrophication as the sub grid-scale ecosystem depositions are used in the critical load calculations. Critical loads will be discussed in a companion paper (Repka et al., 2019). Figure 2d shows that the calculated contributions from BAS shipping in 2016 to depositions of sulphur are very low (Present_Base – Present_NoShip) and much lower than what has been calculated assuming 2014 emissions (Present_HiSulphur – Present_Base) as shown in Figure 2e, with percentage contributions exceeding 10% in many coastal zones.

These findings for the depositions of oxidised nitrogen and sulphur are also illustrated in Table 3 where measured concentrations in precipitation at sites located in the BAS coastal regions are compared to the Present_Base, Present_NoShip and Present_HiSulphur model calculations. Compared to Present_Base, averaged concentrations in precipitation are about 14% lower for oxidised nitrogen when BAS ship emissions are excluded (Present_Base – Present_NoShip). The effects of the

stricter SECA regulations is demonstrated by an increase of about 9% in the calculated concentrations of oxidised sulphur in precipitation in the Present_HiSulphur scenario compared to the Present_Base calculation.

### 3.3 Contributions to individual countries from BAS shipping.

Figure 3 shows the concentrations of $NO_2$, $SO_2$, $PM_{2.5}$, and the depositions of oxidised sulphur and oxidised nitrogen averaged over the individual countries bordering the BAS. The black (Present) and green (Future) bars represent contributions from all other sources (both anthropogenic and natural) than BAS shipping. The blue part of the bars represents the (present and future) contributions from BAS shipping calculated as Base – NoShip where Base can be either Present_Base or Future_Base and NoShip can be either Present_NoShip or Future_NoShip. The sum of the black or blue and the green parts of the bars then adds up to the total concentrations and depositions averaged over the individual countries bordering the BAS for the Present_Base and the Future_Base scenarios. The red part is the additional BAS contributions assuming BAS ship emissions at 2014 levels calculated as Present_HiSulphur – Present_Base. The calculations are made assuming linearity. Previous calculations, adding up contributions from different sources, have shown that this assumption is reasonable (Jonson et al., 2017, 2018). Irrespective of species and depositions, the largest contributions are seen for smaller countries with long coastlines exposed to the BAS as Denmark and the Baltic States, and the least for large countries as Germany and Poland with major parts of their areas located far from the shipping routes.

Following the expected reductions between 2016 and 2030 in both land-based and ship emissions, calculated concentrations and depositions are reduced over the 2016 to 2030 time-span. For $SO_2$ and depositions of sulphur, BAS shipping is already an insignificant source in 2016 and the differences between 2030 and 2016 are almost entirely caused by changes in land-based emissions. For $NO_2$ concentrations and depositions of oxidised nitrogen, reductions of land-based and BAS ship emissions both contribute to the improvements in pollution levels. In the BAS region the fractional reductions of future concentrations attributed to (mainly) land-based, and to BAS ship emissions are roughly in the same range.

The largest contrilbutions from BAS shipping is seen for $NO_2$ (Figure 3b), depositions of oxidised nitrogen (Figure 3c), and partially also for $SO_2$ (Figure 3a) when assuming 2014 emissions (Present_HISulphur). However, for $SO_2$ calculated contributions are insignificant following the implementation of the stricter SECA in 2015. The same conclusion also holds for sulphur depositions (Figure 3d). $PM_{2.5}$ contributions from BAS shipping are markedly smaller than for $NO_2$. Contributions are higher when assuming Present_HiSulphur emissions. After the implementation of stricter SECA regulations in 2015, $PM_{2.5}$ from shipping mainly originates from $NO_2$ and, in part, primary PM emissions. As shown in Figure 1d,e elevated $PM_{2.5}$ concentrations from BAS shipping are mainly seen in coastal zones close to shipping lanes. Much of these coastal zones are densely populated. When assessing the health effects of PM in a forthcoming companion paper (Barregård et al., 2019), population weighted $PM_{2.5}$ concentrations are used.

Figure 4 (left) shows calculated SOMO35[1] as an average for countries around the Baltic Sea and the effect of BAS shipping. The effects on annual average ozone are shown in the same figure (right). For all countries annual averaged ozone is in the 33 - 37 ppb range. For most countries both SOMO35 and annually averaged ozone increase only slightly as a result of BAS

---

[1]SOMO35 is the indicator for health impacts recommended by WHO calculated as the daily maximum of 8-hour running ozone maximum over 35 ppb

shipping, and relatively more so for SOMO35 than for annually averaged ozone. However, in Denmark emissions from BAS shipping result in a decrease in annually averaged ozone with present emissions.

Changes in ozone are caused by a combination of ozone production, mainly in the summer months, and ozone titration by NO, mainly in winter. In winter reductions in $NO_x$ emissions (including reductions in emissions from ships) result in a decrease

in ozone titration and subsequently higher ozone levels. This is illustrated in Figure A5a with ozone winter levels in 2030 higher than in 2016 throughout northern and central Europe. Ozone production dominates in the summer months (Figure A5b) and with the exception of a region around the English channel, the expected reductions in the emissions of ozone precursors result in lower ozone ozone levels. For SOMO35 (Figure A5c,d) the relative increase in winter is much smaller as ozone is largely below the 35 ppb threshold. In summer the increase in ozone from present to future caused by less titration around the English

channel is confined to a much smaller area. As a result annually average ozone production and titration in the BAS region partially cancel out, and for some regions and countries titration dominates the annual values. As shown in Figure 4 (green bars) the expected emission reductions (land based and from ships) from year 2016 to 2030 result in overall reductions in ozone levels (both annually averaged ozone and SOMO35) for all countries except Germany and Denmark, where calculated average ozone levels are higher in 2030 (but SOMO35 is reduced). In 2030 the additional emissions from BAS shipping result

in increased SOMO35 and annually average ozone in all countries. (blue bars in Figure 4). These results are in good agreement with detailed model calculations with projected emission changes, demonstrating a future transition from NMVOC[2]-limited to $NO_x$-limited regimes in large parts of Europe north of the Alps (Beekmann and Vautard, 2010).

It has to be noted that in our model calculations the ship emissions are instantly diluted throughout the model grid cell where the emissions occur. Previous studies Vinken et al. (2011); Huszar et al. (2010) have shown that this could lead to an

overestimation of ozone formation. However, Vinken et al. (2011) found that the overestimation caused by instant dilution was small in polluted regions, such as the central parts of the BAS.

## 4  Conclusions

Our calculations clearly show that, following the stricter SECA regulations from 1 January 2015, sulphur emissions from BAS shipping now contribute little to depositions of oxidised sulphur and $PM_{2.5}$ concentrations in air. This is in contrast to pre-2015

conditions when less stringent sulphur regulations were in place, and even more compared to pre-2011 conditions when up to 1.5% sulphur were allowed in marine fuels in the SECAs.

Still, emissions of $NO_x$ and particles from BAS shipping continue to be high, causing health problems and other detrimental impacts on the environment in the BAS region. At present emission levels, particles originating from BAS shipping are mainly formed from $NO_x$ emissions and partially by primary particles other than $SO_4$.

Currently very little openly available emission factor data exist for marine diesels using Ultra-low sulphur heavy fuel oil and covering the whole engine load range from zero to 100 percent. Hypothetically, with these cases STEAM calculates the $SO_x$ emission factor based on available sulphur in the fuel. If this was close to zero, then the $SO_x$ emission factor is very small. The

---

[2]NMVOC - Non Methane Volatile Organic Compounds

conversion of fuel sulphur to sulphate has a similar mechanism and only a small fraction of available sulphur is converted to $SO_4$. Again, the emission factor for $SO_4$ would be very small if the fuel sulphur content is close to zero. For other species of PM, like EC, OC and Ash, emission factors will be similar as with HFO and thus emissions of non-sulphur particles from BAS shipping are assumed to be virtually unaffected by the SECA regulations.

EMEP source receptor calculations for the individual countries (see EMEP country reports for year 2016 Klein et al. (2018)) show that, for many countries in the region, BAS shipping is among the 5 to 6 largest regions/countries contributing to SIA (Secondary Inorganic Aerosols). SIA is a major constituent of $PM_{2.5}$ typically ranging from about 30 to 60% of $PM_{2.5}$ mass in (scarce) measurements and in EMEP model calculations (Tsyro et al., 2018). Other constituents in $PM_{2.5}$ include seasalt and organics (both natural and anthropogenic) with no or minor contributions from shipping, as well as primary particles. As

a result, the percentage contributions from BAS shipping to SIA is of the order of a factor of two higher than for $PM_{2.5}$. As the natural part of $PM_{2.5}$ (and likewise $PM_{10}$) is not included in the EMEP source receptor calculations (EMEP Status Report 1/2018, 2018) they bears some resemblance to SIA. Thus the relative contributions from BAS shipping presented here is lower than the above source receptor calculations as it is compared to $PM_{2.5}$ (and likewise $PM_{10}$) of both anthropogenic and natural origin. In a global model calculation with ship emission from the BAS and NOS also provided by FMI, source receptor

relationships are in the same range as the reported EMEP results for 2014 and 2016 Jonson et al. (2018). It should however be noted that the EMEP source receptor relationships are calculated by perturbing the emissions by 15%, whereas in this study we have excluded the emissions altogether in the NoShip scenarios.

The largest contributions from shipping are calculated for the coastal zones. Many of the larger cities in the BAS region are located in the coastal zones where contributions can be of the order of of 20% for $NO_2$, but smaller (up to $5-10$%) for $PM_{2.5}$.

In the companion paper (Barregård et al., 2019) health effects from BAS shipping have been adjusted to the population density resulting in a proportionally higher contribution from shipping than presented here as area averaged concentrations.

BAS ship emissions also affect the formation of ground level ozone. In much of the BAS region $NO_2$ levels are already influenced by large land-based sources, and additional contributions from BAS shipping to ozone and ozone metrics, exemplified by SOMO35, is moderate, and for several regions even negative. In this paper we have shown that for most countries future

ozone and ozone metrics are expected to decrease from their present levels.

In addition to influencing particle formation and ozone levels, $NO_x$ emissions also contribute to the depositions of oxidised nitrogen, causing exceedances of critical loads for acidification and in particular eutrophication. Depositions do however depend on the type of landcover. In the EMEP model the calculations of dry depositions are made separately for each sub-grid landcover classification. These sub-grid estimates are aggregated to provide output deposition estimates for broader ecosystem

categories as deciduous and coniferous forests. The ecosystem specific depositions are not shown here, but will be used in a companion paper (Repka et al., 2019) when calculating exceedances of critical loads for acidification and eutrophication.

A significant portion of the depositions of oxidised nitrogen is due to BAS shipping. This is also corroborated by the source-receptor calculations for the individual countries in Europe for 2016, see Klein et al. (2018) where they calculate that BAS shipping is the largest contributor to oxidised nitrogen deposition in Estonia (with 14%), and among the 3 to 5 largest

contributors in several other countries in the region. As discussed above, these depositions are projected to be gradually reduced

following the implementation of the NECA regulations, with relative reductions largely comparable to the decrease from other anthropogenic sources.

Presently there are no further emission mitigation regulations targeted for the Baltic Sea and the North Sea apart from the NECA regulation entering into force in 2021. This regulation is expected to result in gradual reductions in $PM_{2.5}$ concentrations and in depositions of nitrogen from BAS shipping, as shown in our calculations for future versus present conditions. The relative reductions are largely comparable to the decrease from other anthropogenic sources in the region. However, according to (IMO, 2018) the target set by IMO is "to reduce $CO_2$ emissions per transport work, as an average across international shipping, by at least 40% by 2030, pursuing efforts towards 70% by 2050, compared to 2008; and GHG emissions from international shipping to peak and decline as soon as possible and to reduce the total annual GHG emissions by at least 50% by 2050 compared to 2008 whilst pursuing efforts towards phasing them out as called for in the vision as a point on a pathway of $CO_2$ emissions reduction consistent with the Paris Agreement temperature goals." It is unlikely that this goal can be reached without substantial penetration of zero emission ships. If a portion of these zero emission ships run on electricity or hydrogen in 2030 they will be zero emission also for sulphur, nitrogen and $PM_{2.5}$ (in addition to $CO_2$), potentially resulting in reductions of these air pollutants beyond what is assumed in the Future_Base scenario in this paper.

*Code availability.* The EMEP model is available as Open Source (see https://github.com/metno/emep-ctm)

*Data availability.* Model output data available upon request to first author

*Author contributions.* JEJ has made the model calculations and has written most of the paper. MG has assisted in designing the model scenarios and in writing the paper. JPJ and LJ provided the ship emission data for both present and future scenarios. JPJ also assisted in the writing of the paper.

*Acknowledgements.* This work has been funded by European Union (European Regional Development Fund) project EnviSuM, and partially by EMEP under UNECE. Computer time for EMEP model runs was supported by the Research Council of Norway through the NOTUR project EMEP (NN2890K) for CPU, and NorStore project European Monitoring and Evaluation Programme (NS9005K) for storage of data. Surface measurements have been made available through the EBAS web site, http://ebas.nilu.no/Default.aspx, last accessed 27 February 2019.

*Competing interests.* There are no competing interests

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

**Table 1.** All model scenarios have been calculated for the three meteorological years 2014, 2015 and 2016. In the comparisons to measurements in Table 2 only year 2016 model calculations are shown. The land based ECLIPSE emissions for 2016 have been interpolated between 2015 and 2020. SECA regulations for the North Sea are included in the Remaining seas ship emissions. The 2020 sulphur cap is included in the 2030 ship emissions outside the SECAS.

| | Present_Base | Present_NoShip | Present_HiSulphur | Future_Base | Future_NoShip |
|---|---|---|---|---|---|
| Land Based emissions: | ECLIPSE 2016 | ECLIPSE 2016 | ECLIPSE 2016 | ECLIPSE 2030 | ECLIPSE 2030 |
| Baltic Ship emissions: | 2016 | none | 2014 | 2030 | none |
| Remaining Ship emissions: | 2015 | 2015 | 2015 | 2030 | 2030 |

**Table 2.** Annual average measured (Obs) and model calculated concentrations (Calc) of $NO_2$ and $SO_2$ for the present (2016) Base, NoShip, HiSulphur scenarios. The figure continues on the next page with $SO_4$ and $PM_{2.5}$. Also listed are normalized mean bias (NMB), the daily correlations (Corr) and RMS errors (RMS) between model and measurements. For Hallahus there are $PM_{2.5}$ measurements only for parts of the year and bias, correlations and RMS errors are not listed. The timeseries plots for the same sites are shown in appendix A. Km Balt. is a classification of the distance in kilometres between the stations and the Baltic Sea coast. The distance is equal to or smaller than distance listed. The position of the measurement sites and the timeseries plots are shown in appendix A.

## $NO_2$

| Station | Km Balt | Obs | Base Calc | NMB | Corr. | RMS | HiSulphur Calc. | NMB | Corr | RMS | NoShip Calc | NMB | Corr. | RMS |
|---|---|---|---|---|---|---|---|---|---|---|---|---|---|---|
| Aspvreten | 10 | 0.44 | 0.44 | 0.00 | 0.50 | 0.28 | 0.44 | 0.00 | 0.48 | 0.28 | 0.31 | -0.25 | 0.48 | 0.31 |
| Råö | 10 | 1.09 | 1.06 | -0.03 | 0.58 | 0.73 | 0.99 | -0.09 | 0.60 | 0.70 | 0.46 | -0.48 | 0.60 | 0.91 |
| Hallahus | 50 | 0.96 | 0.85 | -0.11 | 0.71 | 0.52 | 0.84 | - 0.12 | 0.71 | 0.52 | 0.58 | -0.40 | 0.70 | 0.64 |
| Anholt | 10 | 1.48 | 0.98 | - 0.34 | 0.73 | 0.96 | 0.92 | -0.38 | 0.76 | 0.99 | 0.35 | -0.76 | 0.66 | 1.55 |
| Keldsnor | 10 | 2.47 | 1.89 | -0.23 | 0.69 | 1.52 | 1.78 | -0.28 | 0.72 | 1.55 | 0.58 | -0.77 | 0.58 | 2.52 |
| Rucava | 100 | 0.75 | 0.38 | -0.49 | 0.63 | 0.56 | 0.38 | -0.49 | 0.63 | 0.56 | 0.30 | -0.60 | 0.57 | 0.63 |
| Zingst | 10 | 2.10 | 0.96 | -0.46 | 0.65 | 1.48 | 0.96 | -0.46 | 0.65 | 1.48 | 0.52 | -0.75 | 0.53 | 1.89 |
| Utö | 10 | 0.95 | 0.57 | -0.40 | 0.76 | 0.58 | 0.59 | -0.38 | 0.76 | 0.56 | 0.17 | -0.82 | 0.25 | 1.00 |

## $SO_2$

| Station | Km Balt. | Obs | Base Calc | NMB | Corr. | RMS | HiSulphur Calc. | NMB | Corr | RMS | NoShip Calc | NMB | Corr. | RMS |
|---|---|---|---|---|---|---|---|---|---|---|---|---|---|---|
| Aspvreten | 10 | 0.10 | 0.25 | 1.50 | 0.11 | 0.34 | 0.30 | 2.00 | 0.13 | 0.38 | 0.24 | 1.40 | 0.11 | 0.34 |
| Råö | 10 | 0.12 | 0.09 | -0.25 | 0.29 | 0.12 | 0.22 | 0.83 | 0.31 | 0.21 | 0.07 | -0.42 | 0.26 | 0.13 |
| Hallahus | 50 | 0.13 | 0.14 | 0.08 | 0.58 | 0.16 | 0.21 | 0.62 | 0.55 | 0.19 | 0.13 | 0.00 | 0.61 | 0.15 |
| Utö | 10 | 0.15 | 0.09 | -0.40 | 0.23 | 0.27 | 0.23 | 0.53 | 0.12 | 0.30 | 0.08 | -0.47 | 0.24 | 0.28 |
| Anholt | 10 | 0.10 | 0.10 | 0.00 | 0.72 | 0.08 | 0.28 | 1.80 | 0.61 | 0.30 | 0.07 | -0.30 | 0.66 | 0.08 |
| Risø | 10 | 0.13 | 0.19 | 0.37 | 0.59 | 0.18 | 0.26 | 1.00 | 0.64 | 0.23 | 0.17 | 0.13 | 0.59 | 0.17 |
| Vilsandi | 10 | 0.30 | 0.11 | -0.63 | 0.37 | 0.43 | 0.18 | -0.40 | 0.28 | 0.42 | 0.10 | -0.67 | 0.38 | 0.43 |
| Zingst | 10 | 0.29 | 0.27 | -0.07 | 0.74 | 0.30 | 0.40 | 0.38 | 0.71 | 0.33 | 0.25 | -0.14 | 0.74 | 0.31 |
| Rucava | 100 | 0.20 | 0.18 | -0.10 | 0.48 | 0.30 | 0.21 | 0.05 | 0.48 | 0.30 | 0.18 | -0.10 | 0.48 | 0.30 |

**Table 2.** continued from previous page.

## SO$_4$

| Station | Km Balt. | Obs | Base | | | | HiSulphur | | | | NoShip | | | |
|---|---|---|---|---|---|---|---|---|---|---|---|---|---|---|
| | | | Calc | NMB | Corr. | RMS | Calc. | NMB | Corr | RMS | Calc | NMB | Corr. | RMS |
| Aspvreten | 10 | 0.71 | 0.56 | -0.21 | 0.74 | 0.48 | 0.65 | -0.08 | 0.72 | 0.49 | 0.56 | -0.21 | 0.74 | 0.49 |
| Råö | 10 | 0.98 | 0.59 | -0.40 | 0.53 | 0.71 | 0.71 | -0.28 | 0.47 | 0.71 | 0.57 | -0.42 | 0.53 | 0.72 |
| Hallahus | 50 | 0.87 | 0.76 | -0.13 | 0.65 | 0.60 | 0.88 | 0.01 | 0.62 | 0.63 | 0.76 | -0.13 | 0.65 | 0.60 |
| Anholt | 10 | 1.58 | 0.60 | -0.62 | 0.62 | 1.16 | 0.73 | -0.54 | 0.58 | 1.08 | 0.59 | -0.63 | 0.62 | 1.18 |
| Risø | 10 | 1.63 | 0.82 | -0.50 | 0.69 | 1.13 | 0.94 | -0.42 | 0.68 | 1.06 | 0.81 | -0.50 | 0.69 | 1.14 |
| Rucava | 100 | 0.92 | 0.80 | -0.13 | 0.71 | 0.64 | 0.88 | -0.04 | 0.71 | 0.63 | 0.80 | -0.13 | 0.71 | 0.65 |

## PM2.5

| Station | Km Balt. | Obs | Base | | | | HiSulphur | | | | NoShip | | | |
|---|---|---|---|---|---|---|---|---|---|---|---|---|---|---|
| | | | Calc | NMB | Corr. | RMS | Calc. | NMB | Corr | RMS | Calc | NMB | Corr. | RMS |
| Hallahus | 50 | 6.04 | 5.90 | -0.02 | | | 6.08 | 0.01 | | | 5.46 | -0.10 | | |
| Aspvreten | 10 | 4.39 | 3.63 | -0.17 | 0.57 | 3.08 | 3.77 | -0.14 | 0.57 | 3.07 | 3.45 | -0.21 | 0.57 | 3.09 |
| Råö | 10 | 3.77 | 4.26 | 0.13 | 0.43 | 3.40 | 4.44 | 0.18 | 0.42 | 3.48 | 3.93 | 0.04 | 0.45 | 3.03 |
| Rucava | 100 | 9.08 | 4.63 | -0.49 | 0.50 | 7.31 | 4.77 | -0.47 | 0.50 | 7.23 | 4.43 | -0.51 | 0.51 | 7.40 |
| Vilsandi | 10 | 4.38 | 3.43 | -0.22 | 0.67 | 3.00 | 3.63 | -0.17 | 0.67 | 2.94 | 3.21 | -0.27 | 0.67 | 3.07 |

**Table 3.** Annual average measured (Obs) and model calculated concentrations (Calc) in precipitation of oxidised nitrogen in mg(N)l−1 and oxidised sulphur in mg(in S)l−1in 2016 for the present Base, NoShip, HiSulphur scenarios. Also listed are the normalized mean bias (NMB), the daily correlations (Corr) and RMS errors (RMS) between model and measurements. Km Balt. is a classification of the distance in kilometres between the stations and the Baltic Sea coast. The distance is equal to or smaller than distance listed. The position of the measurement sites and the timeseries plots are shown in appendix A.

## Wet dep. oxN

| Station | Km Balt. | Obs | Base | | | | HiSulphur | | | | NoShip | | | |
|---|---|---|---|---|---|---|---|---|---|---|---|---|---|---|
| | | | Calc | NMB | Corr. | RMS | Calc. | NMB | Corr | RMS | Calc | NMB | Corr. | RMS |
| Bredkälen | 200 | 0.15 | 0.14 | -0.07 | 0.63 | 0.38 | 0.14 | -0.07 | 0.62 | 0.28 | 0.12 | -0.20 | 0.61 | 0.27 |
| Råö | 10 | 0.55 | 0.80 | 0.45 | 0.57 | 1.21 | 0.80 | 0.45 | 0.57 | 1.21 | 0.72 | 0.31 | 0.57 | 1.15 |
| Preila | 10 | 0.65 | 0.76 | 0.17 | 0.38 | 1.62 | 0.76 | 0.17 | 0.38 | 1.62 | 0.65 | 0.00 | 0.36 | 1.65 |
| Lahemaa | 20 | 0.48 | 0.39 | -0.19 | 0.16 | 0.95 | 0.39 | -0.19 | 0.16 | 0.94 | 0.32 | -0.33 | 0.16 | 0.94 |
| Leba | 10 | 0.73 | 0.78 | 0.07 | 0.59 | 1.05 | 0.78 | 0.07 | 0.59 | 1.04 | 0.67 | -0.08 | 0.53 | 1.10 |

## Wet dep. oxS

| Station | Km Balt. | Obs | Base | | | | HiSulphur | | | | NoShip | | | |
|---|---|---|---|---|---|---|---|---|---|---|---|---|---|---|
| | | | Calc | NMB | Corr. | RMS | Calc. | NMB | Corr | RMS | Calc | NMB | Corr. | RMS |
| Bredkälen | 200 | 0.11 | 0.11 | 0.00 | 0.39 | 0.31 | 0.12 | 0.09 | 0.40 | 0.31 | 0.11 | 0.00 | 0.39 | 0.31 |
| Råö | 10 | 0.23 | 0.40 | 0.74 | 0.54 | 0.66 | 0.45 | 0.96 | 0.55 | 0.70 | 0.40 | 0.74 | 0.53 | 0.65 |
| Preila | 10 | 0.38 | 0.56 | 0.47 | 0.37 | 1.20 | 0.60 | 0.58 | 0.39 | 1.20 | 0.55 | 0.45 | 0.37 | 1.21 |
| Leba | 10 | 0.42 | 0.51 | 0.21 | 0.48 | 0.85 | 0.56 | 0.33 | 0.53 | 0.83 | 0.51 | 0.21 | 0.47 | 0.85 |

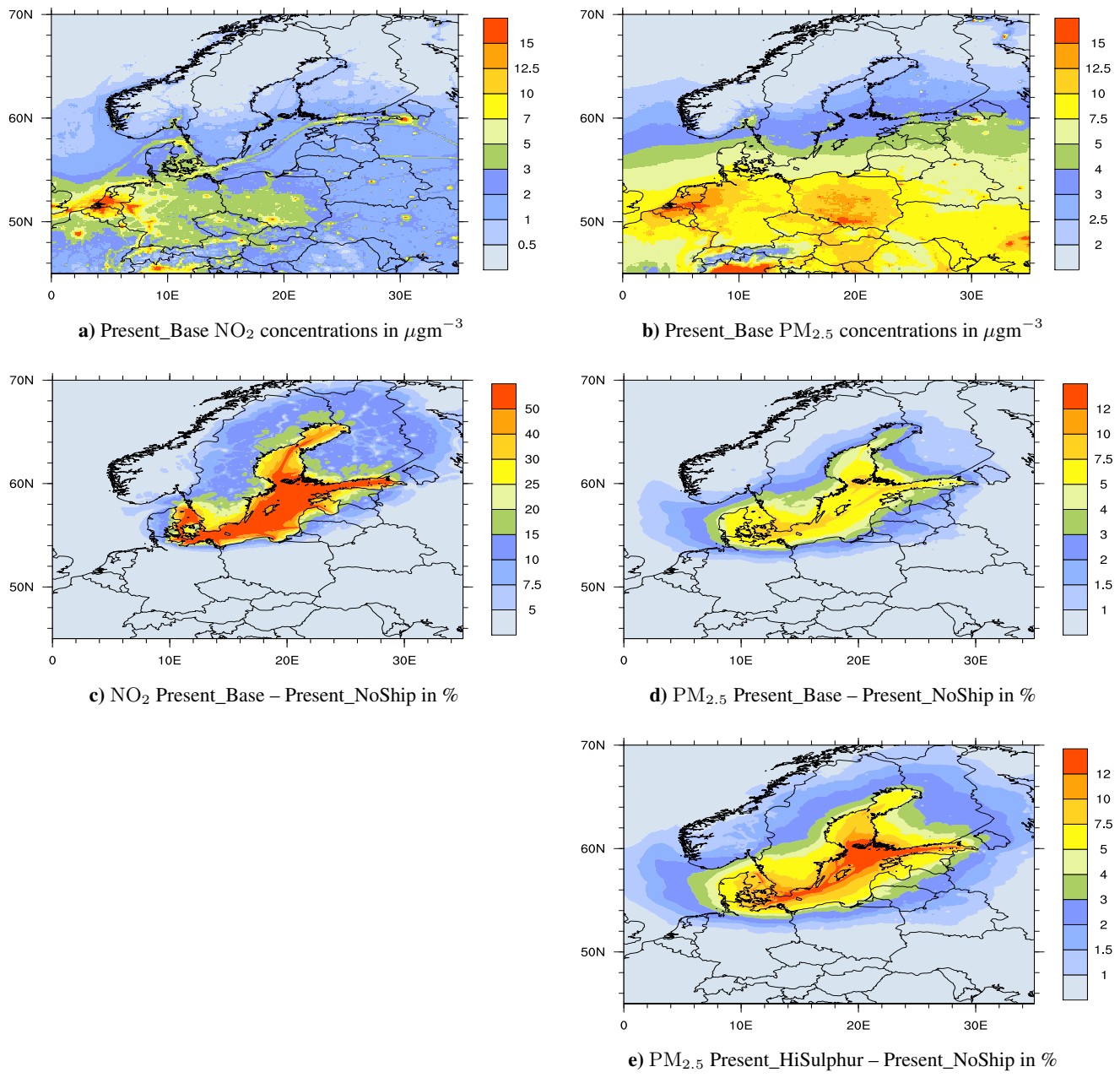

**a)** Present_Base $NO_2$ concentrations in $\mu gm^{-3}$

**b)** Present_Base $PM_{2.5}$ concentrations in $\mu gm^{-3}$

**c)** $NO_2$ Present_Base – Present_NoShip in %

**d)** $PM_{2.5}$ Present_Base – Present_NoShip in %

**e)** $PM_{2.5}$ Present_HiSulphur – Present_NoShip in %

**Figure 1.** Top panels: concentrations of $NO_2$ and $PM_{2.5}$ in the Present_Base case. Middle panels: present percentage contribution from BAS ship emissions to $NO_2$ and $PM_{2.5}$ after the new sulphur regulations. Bottom panel: percentage contribution to $PM_{2.5}$ concentrations before the new sulphur regulations.

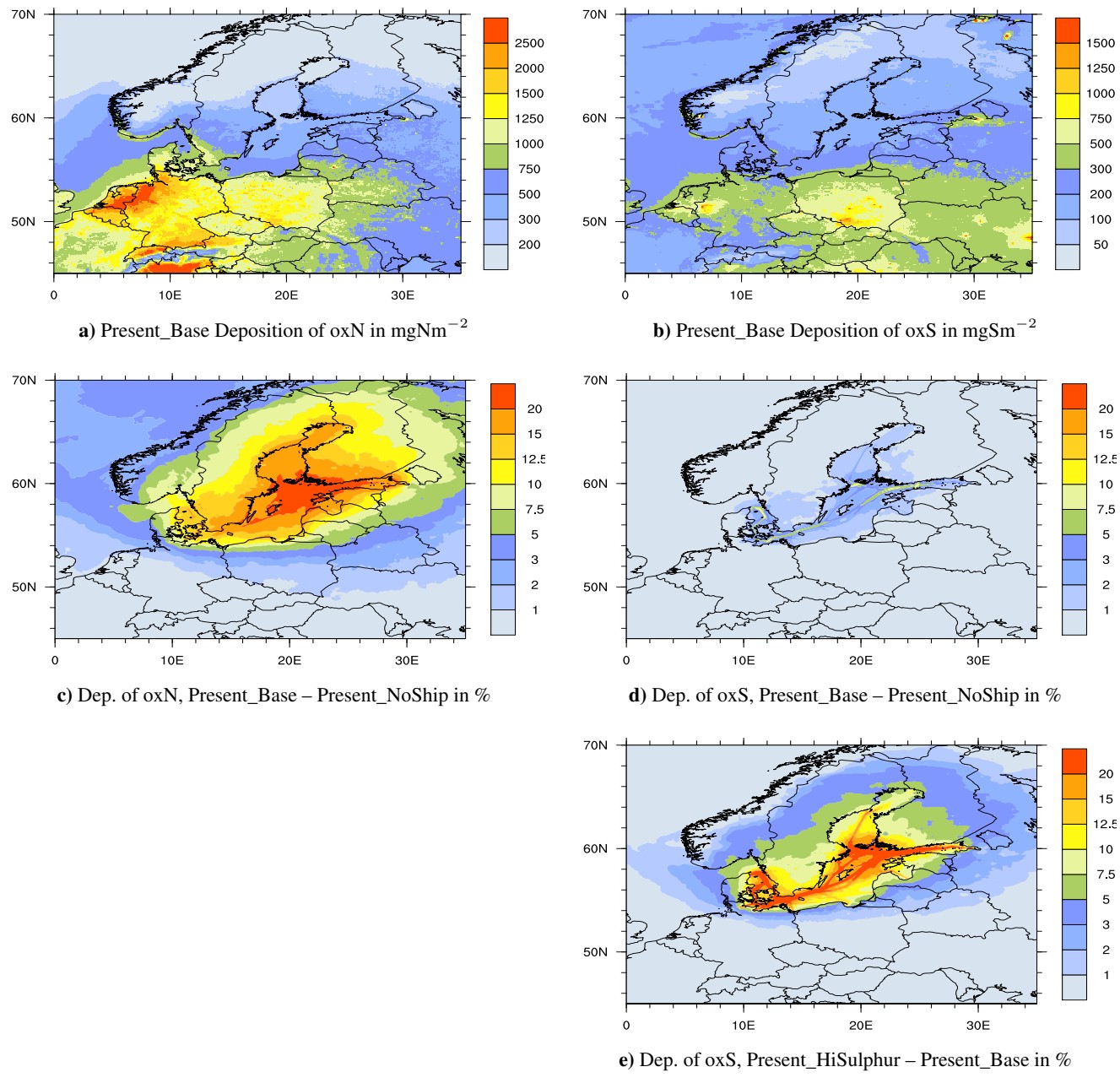

**a)** Present_Base Deposition of oxN in mgNm$^{-2}$

**b)** Present_Base Deposition of oxS in mgSm$^{-2}$

**c)** Dep. of oxN, Present_Base – Present_NoShip in %

**d)** Dep. of oxS, Present_Base – Present_NoShip in %

**e)** Dep. of oxS, Present_HiSulphur – Present_Base in %

**Figure 2.** Top panels: calculated depositions of oxidised nitrogen and sulphur. Middle panels: present percentage contributions from BAS ship emissions to depositions of oxidised nitrogen and oxidised sulphur with reference to Base 2016. Bottom panel: percentage contribution to depositions of oxidised sulphur with reference to 2014 BAS emissions.

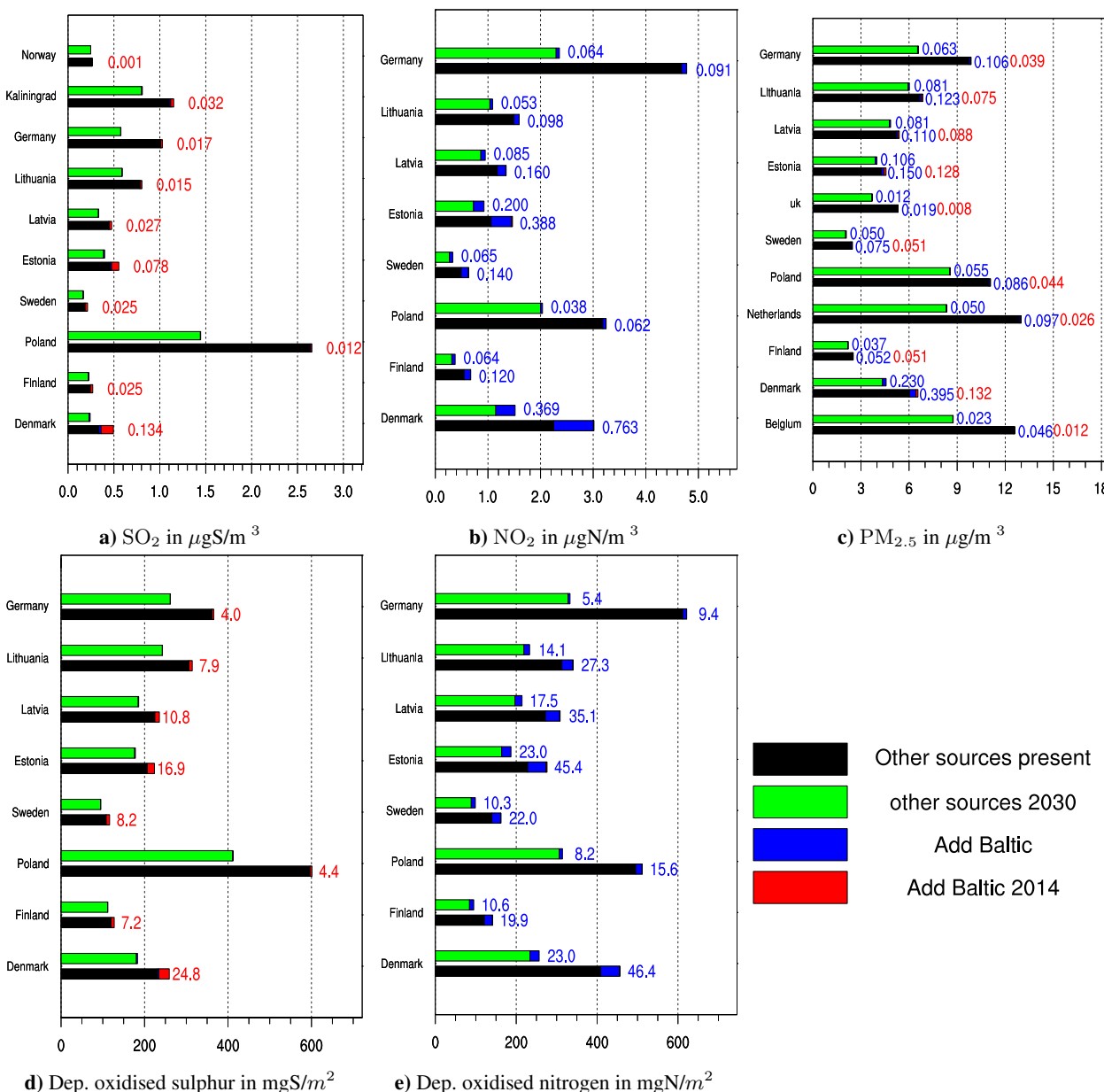

**Figure 3.** For each country, the upper bar shows the future (2030) case and the lower bar the present case country average concentration. a) $SO_2$, b) $NO_2$, c) $PM_{2.5}$, and depositions of oxidised sulphur (d) and oxidised nitrogen (e). The black and green bars represent the Present_NoShip and Future_NoShip calculations respectively. The additional contributions from BAS (Add Baltic) are shown in blue and the additional effect assuming high sulphur fuel emissions (Add Baltic 2014) in red (These are also given as numbers. Numeric values for $NO_2$ Add Baltic and for $SO_2$ Add Baltic 2014 not given as they are very small).

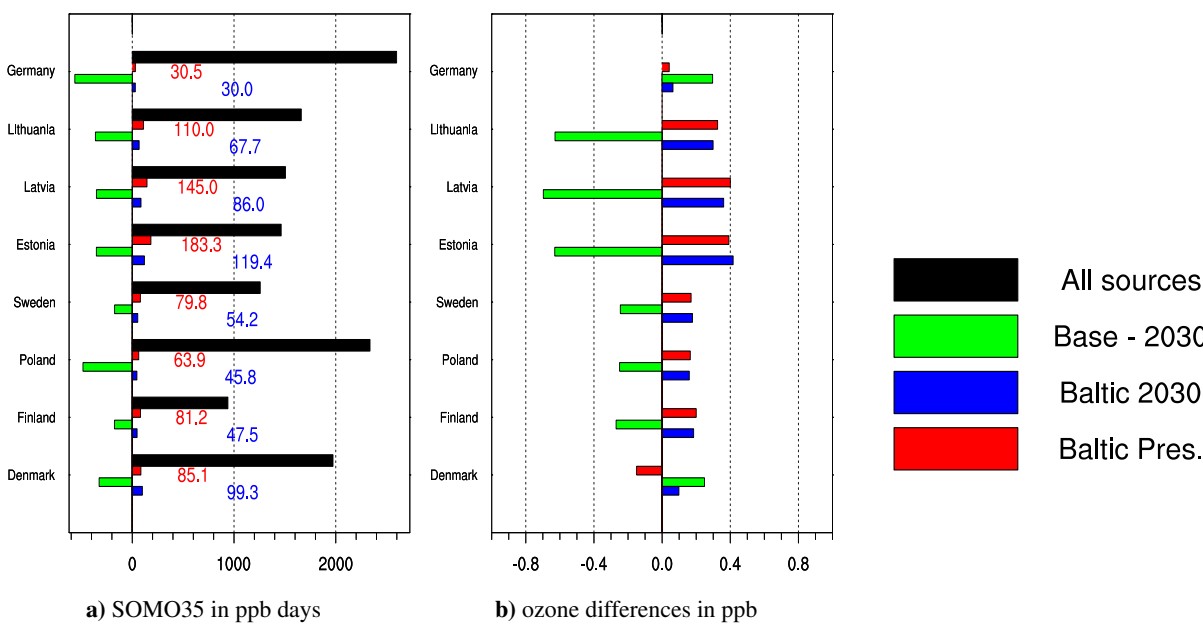

**a)** SOMO35 in ppb days      **b)** ozone differences in ppb

**Figure 4.** Left, SOMO35 in ppb days where black bars represent Present_Base levels. Right, changes in annual ozone in ppb (annual average ozone is in the 30 - 35 ppb range in all countries). For both SOMO35 and annual ozone green bars represent changes in levels from 2016 to 2030 (Present_Base – Future_Base), red bars: contributions from BAS (Present_Base – Present_NoShip) , blue bars: contributions from BAS in 2030 (Future_Base – Future_NoShip).

## Appendix A

This appendix contains time series plots for $NO_2$, $SO_2$ and $PM_{2.5}$ for the meteorological year 2016. Measured and model calculated annual average concentrations, correlations and RMS errors are listed in Table 2 in the main text. For many sites the timeseries for the different model scenarios are virtually identical, and the HiSulphur and NoShip scenarios are masked by the Base scenario.

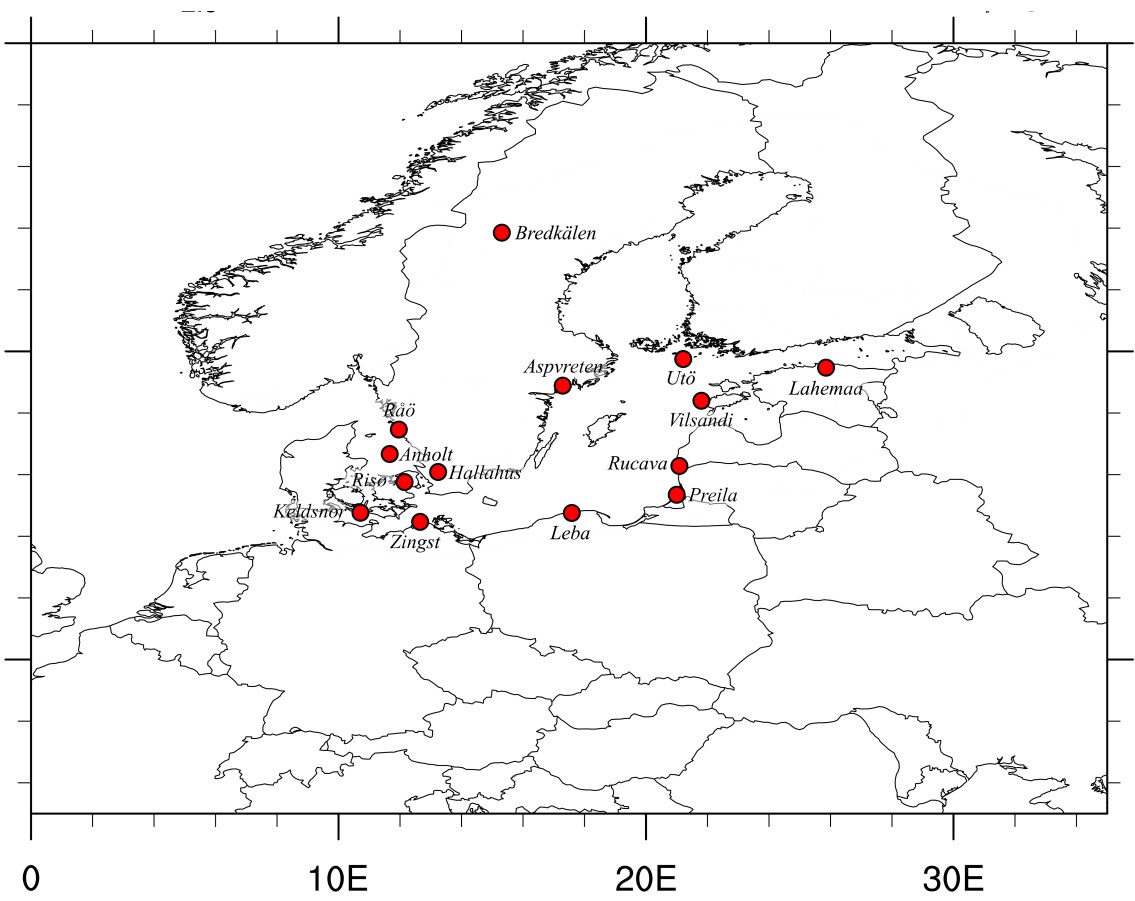

**Figure A1.** Location of the measurement sites shown in Figures A2 to A4 and listed in Tables 2 and 3.

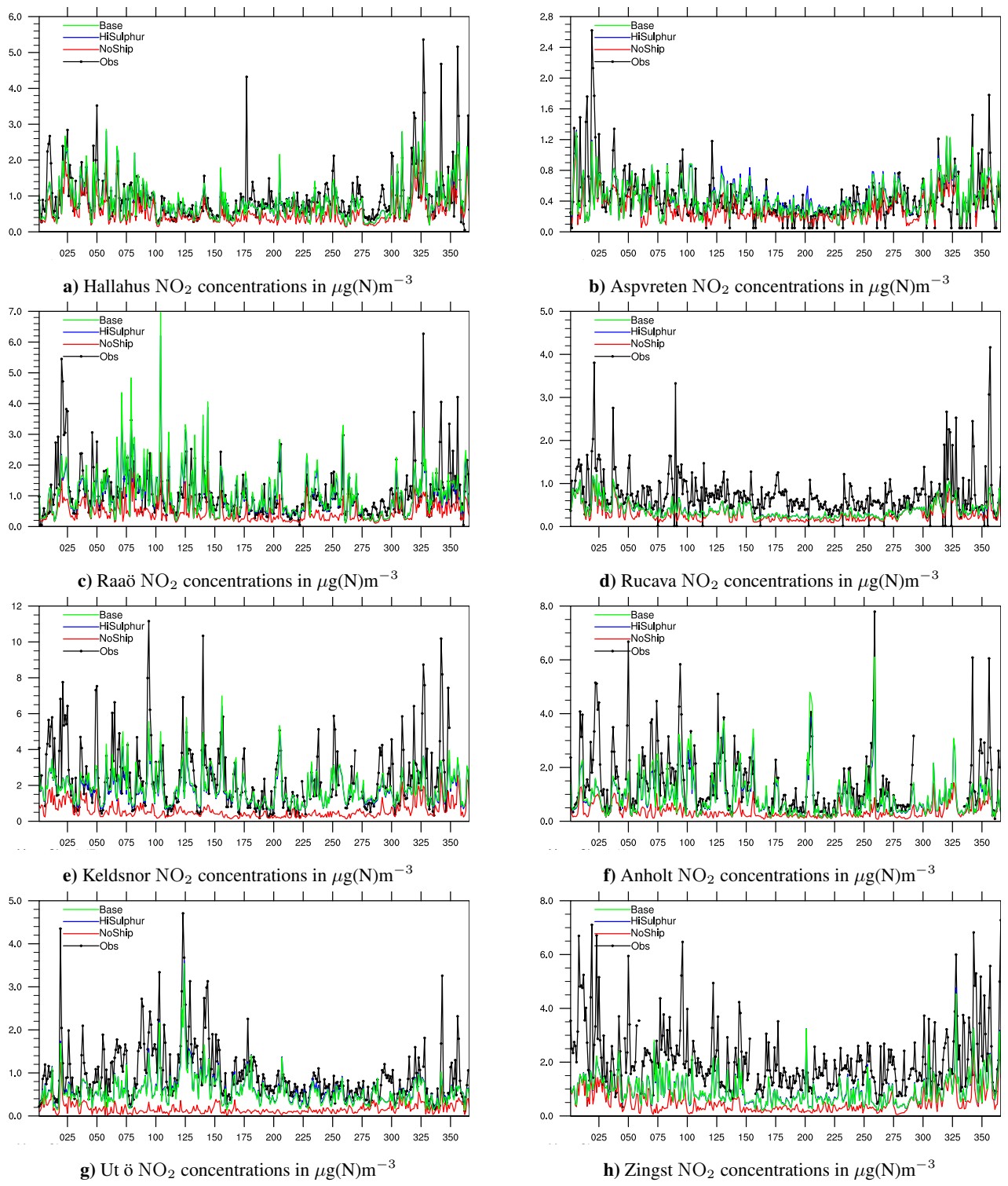

**Figure A2.** Measured and model calculated present (2016) concentrations of NO$_2$. Present model calculated results are shown for the Base HiSulphur and the NoShip scenarios. The HiSulphur calculations are not visible as it is almost identical to Present_Base.

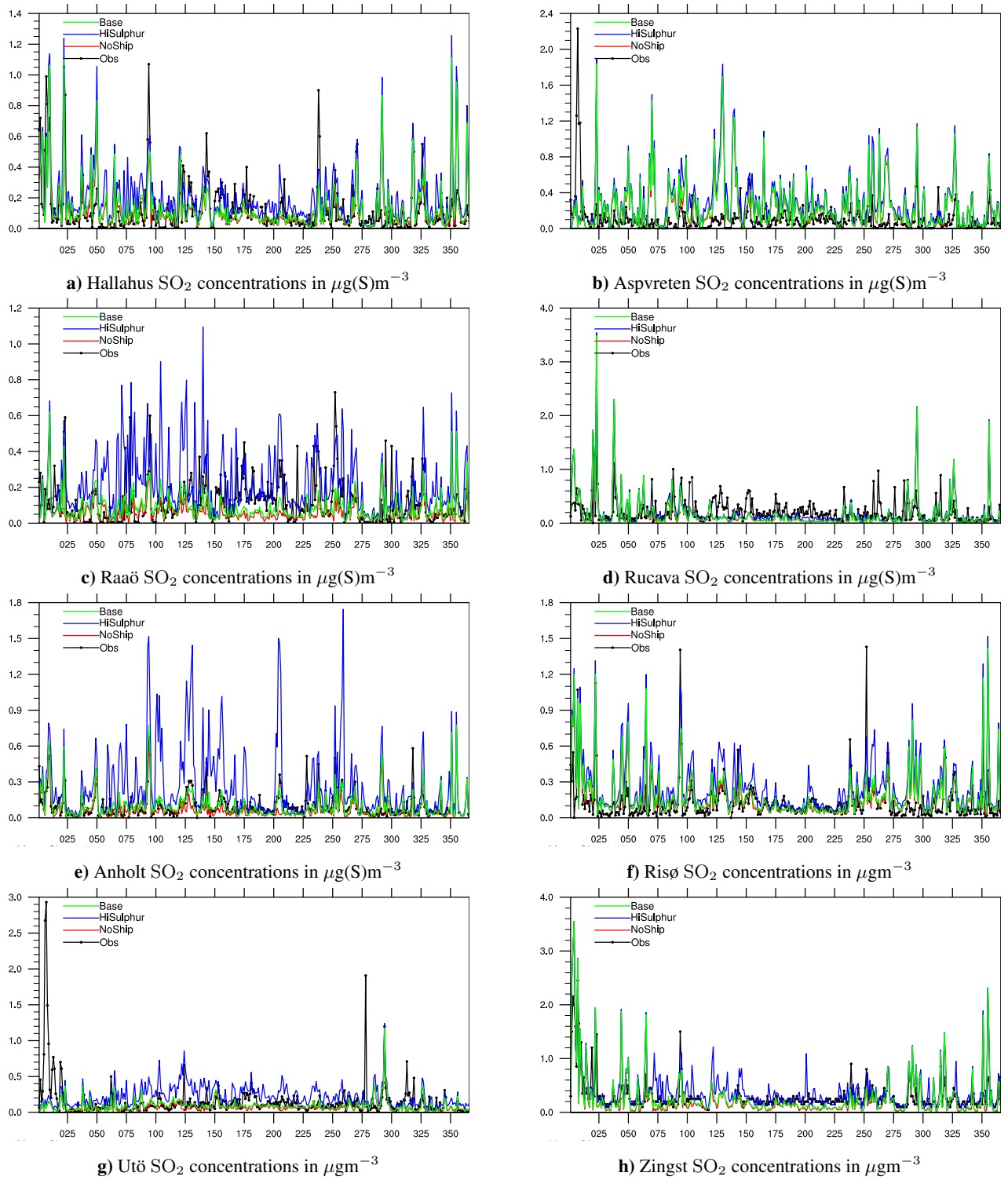

**a)** Hallahus SO$_2$ concentrations in $\mu$g(S)m$^{-3}$

**b)** Aspvreten SO$_2$ concentrations in $\mu$g(S)m$^{-3}$

**c)** Raaö SO$_2$ concentrations in $\mu$g(S)m$^{-3}$

**d)** Rucava SO$_2$ concentrations in $\mu$g(S)m$^{-3}$

**e)** Anholt SO$_2$ concentrations in $\mu$g(S)m$^{-3}$

**f)** Risø SO$_2$ concentrations in $\mu$gm$^{-3}$

**g)** Utö SO$_2$ concentrations in $\mu$gm$^{-3}$

**h)** Zingst SO$_2$ concentrations in $\mu$gm$^{-3}$

**Figure A3.** Measured and model calculated present (2016) concentrations of SO$_2$. Present model calculated results are shown for the Base, HiSulphur and the NoShip scenarios.

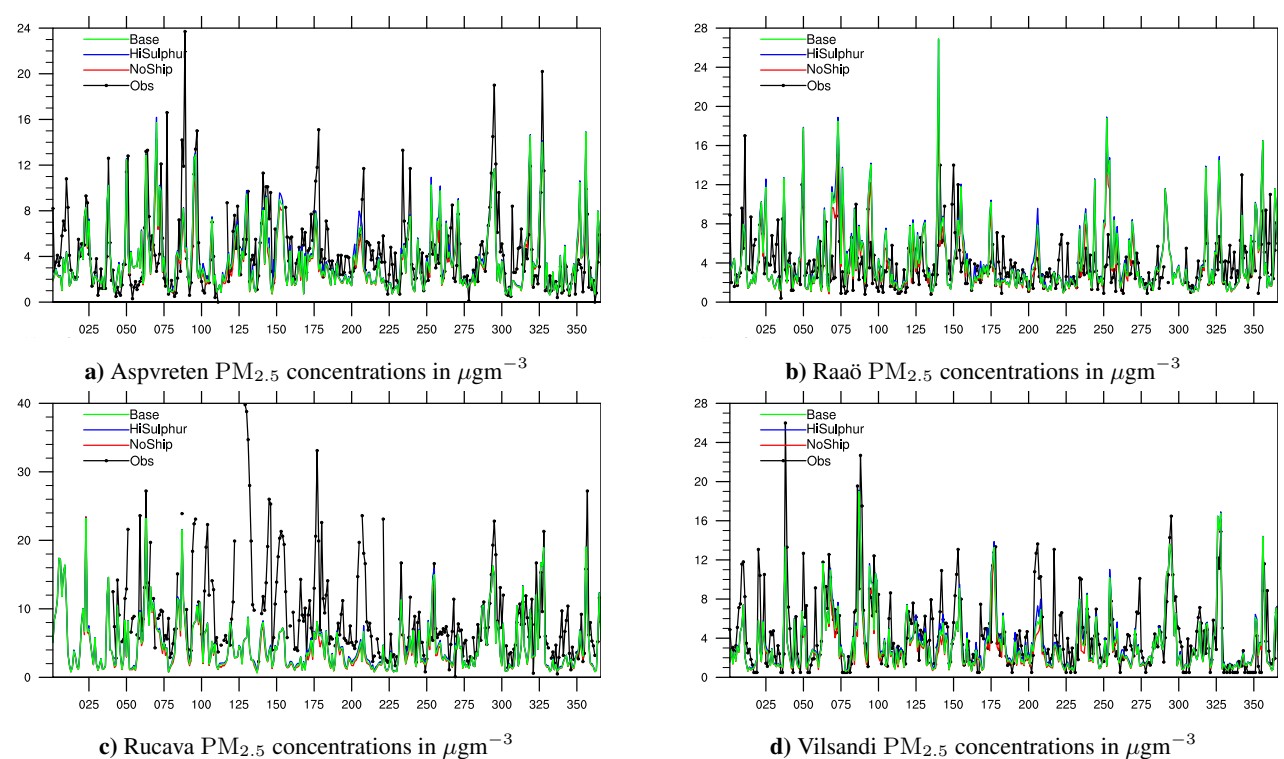

**a)** Aspvreten $PM_{2.5}$ concentrations in $\mu gm^{-3}$

**b)** Raaö $PM_{2.5}$ concentrations in $\mu gm^{-3}$

**c)** Rucava $PM_{2.5}$ concentrations in $\mu gm^{-3}$

**d)** Vilsandi $PM_{2.5}$ concentrations in $\mu gm^{-3}$

**Figure A4.** Measured and model calculated present (2016) concentrations of $PM_{2.5}$. Present model calculated results are shown for the Base, HiSulphur and the NoShip scenarios.

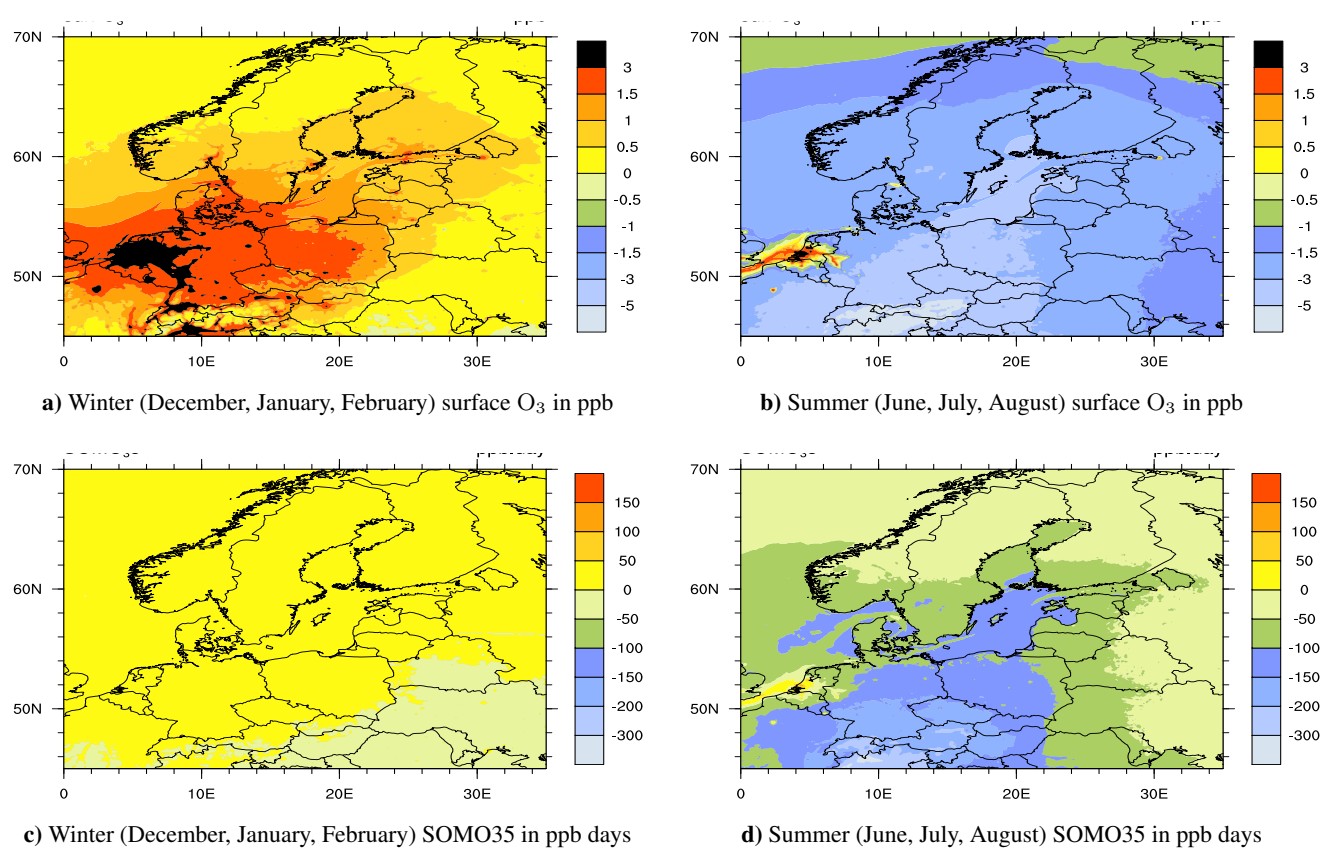

**a)** Winter (December, January, February) surface $O_3$ in ppb

**b)** Summer (June, July, August) surface $O_3$ in ppb

**c)** Winter (December, January, February) SOMO35 in ppb days

**d)** Summer (June, July, August) SOMO35 in ppb days

**Figure A5.** Difference between Future_Base and Present_Base for average surface ozone in Winter (a) and Summer (b) and for SOMO35 in Winter (c) and Summer (d).