# Peer review of "Effects of strengthening the Baltic Sea ECA regulations"

_Atmospheric Chemistry and Physics, 2019_

## Referee Comment (RC1) · Anonymous Referee #1 · 19 Mar 2019

The study presents model calculations with the regional scale EMEP model covering Europe and its adjacent seas with a focus on shipping emissions from the Baltic Sea. The EMEP model is used regularly in the reporting within the frame of the Convention on Long-Range Transboundary Air Pollution. Air pollutant concentrations and deposition from the EMEP model has been evaluated and inter-compared in numerous studies with favorable outcomes. The EMEP model has been used previously in the North Sea and Baltic Sea region for estimating the effect of shipping. Based on the title, I had expected a discussion to which extent the enforced ECA regulations improve air quality in European countries. However, the paper falls short in precisely addressing the effects of the ECA regulations. The paper refers in many places to the use of the presented model results in upcoming studies that are in preparation. To improve

the presentation, I recommend to better emphasize the objectives of this study and the value of the model calculations in itself by deriving recommendations for emission control policies.

The use of three years to compute an average of the present situation is not clear. Information regarding the averaging of computed years is given piecewise and the reader is left alone with finding out which emissions and meteorology of which years are used for the different scenario simulations and which output year is compared. Definitely, a table presenting this information in one place would be very helpful. Why was only one year (2016) compared with the future scenarios?

The non-consistent numbering of sections adds to the confusion: section one starts with the Introduction, followed by a section 'Experimental Setup' which is not numbered and then beginning with 1.1 Emissions. This should probably be section two and renamed 'Model Setup'.

Projections for the future ship emissions are not described and justified in the manuscript. How would the air quality change in future if a higher growth of the ship fleet or non-compliance to the stricter regulations are assumed?

Specific Comments:

1.) P. 2 line 30: Please add a discussion on emissions from open loop scrubbers to air and to water in the Introduction. Moreover, the different alternative fuels and control technologies to fulfil the stricter ECA regulations and their actual use by the BAS ship fleet needs to be addressed.

2.) P.3 line 1-2: At the end of the Introduction, it is referred to two papers in preparation which are based on results of this study. This reference somehow weakens the scientific relevance of the present study. Either delete or move to the Conclusions.

3.) P.3 line 8-9: ECLIPSEv5a: how high is the expected variability of land-based emissions between 2014 and 2016?

4.) P.3 line 18: Which fraction of open loop scrubbers is assumed for BAS shipping emissions in 2014 and in 2016? What is assumed about primary particle emissions from open loop scrubbers?

5.) P.3 line 19-21: Are the total BAS shipping emissions for all other pollutants unchanged between 2014 and 2016?

6.) On P.3 line 17, daily emission grids are introduced. On the same page, line 30-31 it is stated that hourly data was aggregated into monthly ship emissions. The purpose of the daily emission grid remains unclear. How high is the uncertainty of monthly versus hourly emissions when considering the titration of ozone by ship emissions?

7.) P.4 line 6-7: Add reference or delete the sentence on ecosystem specific deposition.

8.) P.5 line 16-23: What is the criterion in this study to conclude that measurements are reproduced by the model, either with or without including ship emissions in the model simulations? The present assessment could be strengthened by use of a quantitative indication for the match between model and measurements.

9.) P. 6 line 1-2: What is the fraction of sulfate in the modelled PM2.5 in 2014 and 2016? If possible, add a comparison of measured and calculated SO4 at the monitoring stations in Table 1.

10.) P. 7 line 1-2: The small national contribution of ship emissions in countries with large in-land area does not really reflect the local significance of this sector. It would be better to calculate the average value in the coastal zone of the countries.

11.) P. 8 line 8: Does the statement about unaffected emissions of non-sulphur particles hold in view of realistic emissions from open loop scrubbers and the PM emissions from burning ultra-low sulfur heavy fuel oil (HFO)? The use of scrubbers might capture a large fraction of PM, not only sulfate.

12.) P. 8 line 20: What is the health impact of negative SOMO35?

Technical Corrections:

P. 5 line 6: The lifetime of NO2 is relatively short.

P. 7 line 19: Please replace "show" by "shown".

Figure 1 and Figure 2: Please add annotation of x- and y-axis (degrees longitude and latitude) around the concentration maps. The plot header lines are partly cut off and not visible.

Figure 3: For some countries the green and red bars are hardly visible. I suggest to add additional plots where the contributions from BAS and from high-sulphur fuel are enhanced.

Figure 4: In figure part (a) cut the x-axis in the plot at 2000 ppb days and add the values for the bars above 2000 inside the plot.

---

## Referee Comment (RC2) · Anonymous Referee #3 · 23 Mar 2019

Jonson et al. presented the effects of ship emissions in the BAS on air quality as well as oxidized nitrogen and sulfur over the countries around BAS. They used a combination of ground measurements and chemical transport modeling to estimate the influence of ship emission on air quality of NO2, PM2.5, and SO2 in the present and in the future 2030. Additionally, the effect of strict SECA on sulfur and PM2.5 concentrations was also investigated too. They showed that Baltic Sea shipping has largest impact on NO2 concentration over the central Baltic Sea as well as the coastal cities. The effects of current ship emissions are also shown in PM2.5 and deposition of oxidized nitrogen and sulfur. In terms of contribution of present and future ship emissions on species and depositions over the countries surrounding the BAS, they found that the effects of ship emission are larger in some countries.

Overall this study provides an investigation of BAS ship emissions on air quality and deposition over the BAS and the countries surrounding it and highlights the importance of ship emissions in BAS. However, this paper has some important points that have to be clarified or addressed.

Specific comments:

1. Page 3, line 10: Please specify what FMI stands for.

2. Page 3, line 17: Here the ship emissions in daily temporal scale was first introduced, but later in line 30, it was mentioned that ship emission data in hourly temporal resolution was aggregated into monthly resolution in the CTM. Please clarify the original temporal resolution of ship emissions and how it was implemented into the CTM.

3. Page 3, line 18: What is the spatial resolution of ship emissions?

4. Page 3, line 24-29: The description of future emission projections for the year 2030 is not clear. Although it was mentioned some changes such as vessel size growth and fleet size increase, it will be helpful to include the exact or estimated percentage of ship size growth in 2030 compared to current ship size.

5. Page 4, line 18: I understand the authors mainly focused on the influence of ship emissions, so they presented their results by averaging the three meteorological years. However, some important messages were missing if they took this approach. For example, the changes of observed $SO_2$ and associated $PM_{2.5}$ species before and after stricter SECA regulation was applied, which is highly relevant to the title of this paper and has critical policy implication. I suggest the authors add the comparison of observation between years to the paragraphs where the Present_HiSulphur and Present_Base simulation was compared.

6. Page 5, line 12: As it showed that the elevated $PM_{2.5}$ by BAS shipping emissions are concentrated over the shipping routes and the coastal cities close to the routes, the location of the cities relative to major shipping routes is important. It will be helpful

to include a map showing the geographic location of the sites in Table 1 and 2. Also a discussion about whether the sites close to major shipping route would have larger impact of ship emission can be included.

7. Page 5, line 28: What are differences between your estimated contribution of PM2.5 by BAS shipping emissions and the estimation by Karl et al. (2019) that mentioned in page 2, line 24?

8. Page 5, line 30: The results showed that ship emissions contributed more on PM2.5 concentrations when the ship emissions were assumed to be at 2014 levels. Does it imply the PM2.5 contribution in 2014 (before strict SECA) was mainly from SOx? What is the fraction of sulfate in the modeled PM2.5 in Present_HiSulphur and Base simulation? Do the ground observations of PM2.5 show higher fraction of sulfate in 2014 and decreased fraction of sulfate in 2016 after the strict SECA?

9. Page 5, line 32: Here you compared the differences between Present_Base and Present_Noship for NO2 and PM2.5 at the measurement sites. As the magnitude of NO2 and PM2.5 are different (it is not appropriate to compare their difference directly) and it is hard to tell the differences by eyeballing the numbers, I suggest to have barplots over a map (i.e. every site has its relative difference of Base and Noship (Base minus Noship divided by Noship) for PM2.5 and NO2 presented by a barplot), if it would not be too messy on a map.

10. Page 5, line 34 & Page 6, line 1-2: It was stated that the model results underestimate the measurement at most of the sites listed. What is the criterion of evaluating Base model performance?

11. Page 6, line 33: This discussion would benefit from a quantitative indication than just describing the largest contribution are seen for smaller countries with long coastline. I suggest to add a quantitative assessment like the contribution weighted by coastline length or weighted by distance-to-major-shipping-route to strengthen the statement here.

12. Page 7, line 6: Figure 3b shows the reduction of NO2 caused by ship emission in 2030 (i.e. For each country, the green bar along with blue bar is shorter than the green bar with black bar). As it is stated here, the improvement of the pollution levels is caused by reduction of BAS ship emissions. However, in page 3, line 24-29, you mentioned increase of ship size and fleets, and in page 4, line 23, the future scenario was assumed with NECA (and strict SECA?) applied. How the vessel size growth and fleet size increase, which would lead to more emissions, are balanced with strict regulations to have emission reduction in the future?

13. Page 7, line 21: It was stated that increase of SOMO35 is more than annually averaged ozone. The units of SOMO35 and annually averaged ozone are different. What is the comparing criterion in this statement?

14. Page 7, line 23 & line 24: It was explained the changes in SOMO35 and annually averaged ozone by combination effect of ozone enhancement in the summer and decrease during the winter time. It would be supportive to add analysis of separating SOMO35 and ozone difference by two seasons into appendix and references to support the argument.

15. Page 7, line 25: There are some confusion for the discussion here. In Figure 4, both Germany and Denmark show decrease of annual mean ozone in 2030 (Present-2030, positive difference), but the statement here is "the additional emissions from BAS shipping lead to 'reductions' in annual ozone in Denmark. Furthermore, ….. result in 'increased' annual ozone levels in Germany." Conflict arises from the differences between discussion mentioned above and Figure 4.

16. Page 7, line 27: It is not clear in the discussion here. Figure 4 shows that the SOMO35 increases in the future (also stated in Page 7, line 21) for the two cases, but the statement here – "Even though annual ozone….. lower emissions will result in SOMO35 'reductions' in both these two cases…..". – you mentioned 'reduction' in SOMO35 instead. Additionally, I didn't see the clear connection between SOMO35

reduction and winter titration events.

Technical comments:

1. Page 7, line 9: It should be Figure 3e, instead of Figure 3d.

2. Page 7, line 19: Please rewrite the sentence, "Also show are the effects. . ..".

3. Figure 1: Please add X-axis and Y-axis label of longitude and latitude and remove the remaining cut-off headers in the plots.

---

## Author Comment (AC1) · 28 May 2019

We, the authors, thank the reviewers for constructive comments and suggestions

Below we list the comments from reviewer 1 followed by our reply reference to any changes made in the paper.

General comments

1) However, the paper falls short in precisely addressing the effects of the ECA regulations. The paper refers in many places to the use of the presented model results in upcoming studies that are in preparation. To improve the presentation, I recommend to better emphasize the objectives of this study and the value of the model calculations in itself by deriving recommendations for emission control policies.

[Figure]

Answer: We have added new material to the conclusions reflecting these comments.

Presently there are no further emission mitigation regulations targeted for the Baltic Sea and the North Sea apart from the NECA regulation entering into force in 2021. This regulation is expected to result in gradual reductions in PM2.5 concentrations and in depositions of nitrogen from BAS shipping, as shown in our calculations for future versus present conditions. The relative reductions are largely comparable to the decrease from other anthropogenic sources in the region. However, according to IMO (2018) the target set by IMO is "to reduce $CO_2$ emissions per transport work, as an average across international shipping, by at least 40% by 2030, pursuing efforts towards 70% by 2050, compared to 2008; and GHG emissions from international shipping to peak and decline as soon as possible and to reduce the total annual GHG emissions by at least 50 % by 2050 compared to 2008 whilst pursuing efforts towards phasing them out as called for in the vision as a point on a pathway of $CO_2$ emissions reduction consistent with the Paris Agreement temperature goals." It is unlikely that this goal can be reached without substantial penetration of zero emission ships resulting in reductions of all air pollutants beyond what is assumed in the Future_Base scenario in this paper.

2) The use of three years to compute an average of the present situation is not clear. Information regarding the averaging of computed years is given piecewise and the reader is left alone with finding out which emissions and meteorology of which years are used for the different scenario simulations and which output year is compared. Definitely, a table presenting this information in one place would be very helpful. Why was only one year (2016) compared with the future scenarios?

Answer: As suggested we have included a table listing what emissions have been used in the model scenarios. The effects of future scenarios were calculated for all three meteorological years.

3) The non-consistent numbering of sections adds to the confusion: section one starts

with the Introduction, followed by a section 'Experimental Setup' which is not numbered and then beginning with 1.1 Emissions. This should probably be section two and renamed 'Model Setup'.

Answer: The section numbering has been changed.

4) Projections for the future ship emissions are not described and justified in the manuscript. How would the air quality change in future if a higher growth of the ship fleet or non-compliance to the stricter regulations are assumed?

Answer from Jukka Pekka Jalkanen:

We (Finnish Meteorological Inst.) are preparing a separate manuscript for scenario development and we wanted to keep this part of the manuscript relatively simple, because this is a long story of its own. The key idea is that a simple scaling up emissions with assumed annual growth rate will not work for future years if energy efficiency gains, future emission regulations, fleet technology developments and regional rules are not properly covered. In this regard, we have divided the scenario development in three parts, which will operate on different ship types in a different way. The three features listed in the manuscript involve a) energy efficiency developments, b) vessel size development and c) vessel numbers. These three contributions are used linking the shipping sector to global transport demand, which in turn is linked to annual GDP growth of various regions in the world.

Efficiency gains for various ship types are obtained from Kalli et al. (2013) (see reference in paper). Vessel numbers of each ship type are based on number of ships built each year. For some vessel types this is very challenging, like for the global containership sector, which has undergone a rapid growth since year 2000. The future shipping fleet is difficult to predict, because for example plotting the number of containerships built each year leads to almost exponential growth which cannot be followed for the next 30 years. In 2050, the number of containerships fleet is assumed to grow by 40%, to 6500 actively used ships. Also, the size development of vessels in various

Interactive
comment

shipping sectors needs to be considered. The largest Triple E class of container ships in 2014 were able to carry over 18Âă000 TEUs, but 22Âă000 TEU capacity was nearly achieved in 2017. The triple E class has DWT/TEU ratio of almost 11 tons/TEU, which leads to significantly larger vessels if the current DWT growth trend continues. Containership which exceed 50Âă000 TEUs could be introduced to the fleet by 2050 with our assumptions, which is a decade sooner than anticipated in some recent estimates (McKinsey Group, 2017). These vessels will not be operated in the Baltic Sea because of several limitations.

Specific Comments:

1.) P. 2 line 30: Please add a discussion on emissions from open loop scrubbers to air and to water in the Introduction. Moreover, the different alternative fuels and control technologies to fulfil the stricter ECA regulations and their actual use by the BAS ship fleet needs to be addressed. From 2014 to 2016 only the sulphate fraction of PM was reduced accordingly whereas other components of PM were less affected.

Answer: This discussion is now included in the introduction.

2.) P.3 line 1-2: At the end of the Introduction, it is referred to two papers in preparation which are based on results of this study. This reference somehow weakens the scientific relevance of the present study. Either delete or move to the Conclusions.

Answer: These references have been moved to the conclusions

3.) P.3 line 8-9: ECLIPSEv5a: how high is the expected variability of land-based emissions between 2014 and 2016?

Answer:

In this paper we use the ECLIPSE emissions available only on 5 year intervals.We then apply the same Eclipse emissions for all three meteorological years. We use the ECLIPSE emissions in order to get consistent available emissions for both present and 2030 conditions.

However, annually reported emissions for all countries in Europe are listed in the EMEP reports (reference added in the paper). We have added the following text in the paper:

"In reality land based emissions will change between years. Annual emissions from year 2000 to 2016 for the European countries are listed in EMEP (2018). In the Baltic region reported changes in country emission are small with the exception of SOx emissions in Poland dropping almost 20% from 2014 to 2016."

In the paper we deliberately did not change land-based emissions from year to year in order to isolate the effect of the regulations on shipping.

4.) P.3 line 18: Which fraction of open loop scrubbers is assumed for BAS shipping emissions in 2014 and in 2016? What is assumed about primary particle emissions from open loop scrubbers?

Answer: This information is now included in the paper.

"Globally, during 2014 there were 77 vessels using a scrubber, of which 30% were of open loop, 48% of closed loop and 22% of hybrid type. By 2016 scrubber installations were doubled globally to 155 units. In the Baltic Sea area during 2016, there were 85 vessels operating a scrubber releasing 73 million tonnes of wash water to the sea. Almost all of this (99.8\%) discharge came from open loop operation of scrubbers. "

5.) P.3 line 19-21: Are the total BAS shipping emissions for all other pollutants unchanged between 2014 and 2016?

Answer: Ship emissions of other species differ between 2014 and 2016, but much less than for sulphur.

The following text is now included: "Ship emitted pollutants were modelled using AIS data for year 2014 and 2016. Any changes in vessel activity, fleet size and development will have an impact on energy use and all pollutant emissions. However, the sulphur rule was the only significant change which had a large impact on emitted pollutants. Both PM and SOx were reduced by this change, but only the sulphate fraction of PM

was reduced accordingly whereas other components of PM were less affected."

6.) On P.3 line 17, daily emission grids are introduced. On the same page, line 30-31 it is stated that hourly data was aggregated into monthly ship emissions. The purpose of the daily emission grid remains unclear. How high is the uncertainty of monthly versus hourly emissions when considering the titration of ozone by ship emissions?

Answer: We have corrected the text from daily to hourly. Previously we have run the model with daily ship emissions resulting in only small changes hardly affecting the model validation at measurement sites.

7.) P.4 line 6-7: Add reference or delete the sentence on ecosystem specific deposition.

Answer: Moved to conclusions. References included here.

8.) P.5 line 16-23: What is the criterion in this study to conclude that measurements are reproduced by the model, either with or without including ship emissions in the model simulations? The present assessment could be strengthened by use of a quantitative indication for the match between model and measurements.

Answer: A quantitative indication is given in Table 1 in the form of correlation, rmse and now also bias. There is no commonly accepted threshold for when a model performs well, and it is clear that models (and also emission inventories) often have problems in reproducing a short-lived species such as $NO_2$ correctly. But the point with this paragraph was that the (mainly negative) biases in the model become considerably worse (more negative) at all measurement sites when omitting the ship emission source. This is a clear indication of the importance of the ship emission source of $NO_x$ at these coastal sites. Likewise for $SO_2$ the positive bias becomes very large for the sites listed when the 2016 emissions are replaced by 2014 emissions in the Baltic Sea. For secondary species ($SO_4$ and PM2.5) and depositions of oxidized N and S the effects of shipping is smaller, and we can't draw any conclusions from the match between model and measurement alone with regard to the effects of Baltic Sea emissions.

9.) P. 6 line 1-2: What is the fraction of sulfate in the modelled PM2.5 in 2014 and 2016? If possible, add a comparison of measured and calculated SO4 at the monitoring stations in Table 1.

Answer: SO4 now included in Table 2

We have also included some additional text here to explain the results: In Table 2 we also show measured and model calculated concentrations of SO4. At the sites in Table 2 both the measured and model calculated fraction of SO4 in PM2.5 is about 0.15, and fraction increase only marginally with the Present_HiSulphur scenario.

10.) P. 7 line 1-2: The small national contribution of ship emissions in countries with large in-land area does not really reflect the local significance of this sector. It would be better to calculate the average value in the coastal zone of the countries.

Answer: In this paper we have used similar methodology as used in the source-receptor calculations in the annual EMEP reports (https://www.emep.int/mscw/mscw_publications.html). In the paper the effects along the Baltic Sea coast is also shown in Figures 1 and 2. In the Barregård et al. paper now submitted to IJERPH population weighted concentration are used.

11.) P. 8 line 8: Does the statement about unaffected emissions of non-sulphur particles hold in view of realistic emissions from open loop scrubbers and the PM emissions from burning ultra-low sulfur heavy fuel oil (HFO)? The use of scrubbers might capture a large fraction of PM, not only sulfate.

Answer: We have added a section in the conclusions discussing this:

"For other species of PM, like EC, OC and Ash, emission factors will be similar as with HFO and thus emissions of non-sulphur particles from BAS shipping are assumed to be virtually unaffected by the SECA regulations.

12.) P. 8 line 20: What is the health impact of negative SOMO35?

Answer: Decreases in SOMO35 (caused by increased NOx resulting in ozone titration) should have a positive health impact. However, the corresponding increase in NOx will, as we demonstrate in the paper, increase PM2.5. As PM2.5 has larger effects on health than ozone the net health effects from NOx should still be negative.

Technical Corrections:

P. 5 line 6: The lifetime of NO2 is relatively short.

Answer: Changed to relatively short.

P. 7 line 19: Please replace "show" by "shown".

Answer: This part of thext is changed as a result of comments from reviewer 3.

Figure 1 and Figure 2: Please add annotation of x- and y-axis (degrees longitude and latitude) around the concentration maps. The plot header lines are partly cut off and not visible.

Answer: Figures 1 and 2 changed as requested. Figure 2a showed total (oxidised + redused) depositions of N. Corrected to oxidised N.

Figure 3: For some countries the green and red bars are hardly visible. I suggest to add additional plots where the contributions from BAS and from high-sulphur fuel are enhanced.

Answer: Instead of making additional plots we have added the values for the small "Add Baltic" and "Add Baltic 2014" as numbers behind the bars. We have also changed the colour codes to make the text more visible. We believe that these changes make the charts more readable.

Figure 4: In figure part (a) cut the x-axis in the plot at 2000 ppb days and add the values for the bars above 2000 inside the plot.

Answer: We have added the values for the smaller red and blue bars inside the plot.

---

## Author Comment (AC2) · 28 May 2019

We, the authors, thank the reviewers for constructive comments and suggestions

Below we list the comments from reviewer 1 followed by our reply reference to any changes made in the paper.

Specific comments: 1. Page 3, line 10: Please specify what FMI stands for.

Answer: Finish Meteorologcal Institute now included in brackets.

2. Page 3, line 17: Here the ship emissions in daily temporal scale was first introduced, but later in line 30, it was mentioned that ship emission data in hourly temporal resolution was aggregated into monthly resolution in the CTM. Please clarify the original

temporal resolution of ship emissions and how it was implemented into the CTM.

Answer: The original temporal resolution is hourly. This is corrected in the text.

3. Page 3, line 18: What is the spatial resolution of ship emissions?

Answer: Spatial resolution for the ship emissions For the 2016 Baltic Sea emissions is about 0.034 x 0.018 degrees

For all other sea areas (2015) the spatial resolution is 0.09 x 0.089 degrees

The resolution of the ship emissions is finer than the model grid. Ship emissions are read into the model in the original spatial resolution and then interpolated to the model grid on the fly.

4. Page 3, line 24-29: The description of future emission projections for the year 2030 is not clear. Although it was mentioned some changes such as vessel size growth and fleet size increase, it will be helpful to include the exact or estimated percentage of ship size growth in 2030 compared to current ship size.

Answer: We now list the annual ship size growth used in the 2030 scenario compared to current ship size.

5. Page 4, line 18: I understand the authors mainly focused on the influence of ship emissions, so they presented their results by averaging the three meteorological years. However, some important messages were missing if they took this approach. For example, the changes of observed SO2 and associated PM2.5 species before and after stricter SECA regulation was applied, which is highly relevant to the title of this paper and has critical policy implication. I suggest the authors add the comparison of observation between years to the paragraphs where the Present_HiSulphur and Present_Base simulation was compared.

Answer: The comparisons of model calculations to observations are discussed in section 3.1, where also the Present_HiSulphur and Present_Base simulations are discussed.

6. Page 5, line 12: As it showed that the elevated PM2.5 by BAS shipping emissions are concentrated over the shipping routes and the coastal cities close to the routes, the location of the cities relative to major shipping routes is important. It will be helpful to include a map showing the geographic location of the sites in Table 1 and 2. Also a discussion about whether the sites close to major shipping route would have larger impact of ship emission can be included.

Answer: A map with the position of the measurement sites is now included in the appendix. In the text we also note that the sites close to major shipping routes (as Anholt and Raaoe) NO2 and SO2 measurements can only be reproduced in the Present_Base calculation.

7. Page 5, line 28: What are differences between your estimated contribution of PM2.5 by BAS shipping emissions and the estimation by Karl et al. (2019) that mentioned in page 2, line 24?8.

Answer: Unfortunately there is very little overlap in the stations (even though the AIR-BASE dataset also includes the EMEP sites used in this publication. Although the model resolution is the same the Karl et al. calculations have been made with an older EMEP version. Furthermore also land based emissions are lower in 2016 compared to 2012, especially of SO2. In the present study there is a tendency for more underestimation of NO2 and comparable results for the other species.

Page 5, line 30: The results showed that ship emissions contributed more on PM2.5 concentrations when the ship emissions were assumed to be at 2014 levels. Does it imply the PM2.5 contribution in 2014 (before strict SECA) was mainly from SOx? What is the fraction of sulfate in the modeled PM2.5 in Present_HiSulphur and Base simulation? Do the ground observations of PM2.5 show higher fraction of sulfate in 2014 and decreased fraction of sulfate in 2016 after the strict SECA?

Answer: measured and model calculated SO4 is now tabulated in the paper. The fraction of SO4 in PM2.5 was higher in 2014, but it was not the main component in PM2.5.

9. Page 5, line 32: Here you compared the differences between Present_Base and Present_Noship for NO2 and PM2.5 at the measurement sites. As the magnitude of NO2 and PM2.5 are different (it is not appropriate to compare their difference directly) and it is hard to tell the differences by eyeballing the numbers, I suggest to have barplots over a map (i.e. every site has its relative difference of Base and Noship (Base minus Noship divided by Noship) for PM2.5 and NO2 presented by a barplot), if it would not be too messy on a map.

Answer We have tried this, but given the format of the maps and the plotting software at hand it turned out not to be feasible without compromising readability.

10. Page 5, line 34 & Page 6, line 1-2: It was stated that the model results underestimate the measurement at most of the sites listed. What is the criterion of evaluating Base model performance?

Answer: The criterion is based on the comparison to measurements. This is discussed in more detail in the EMEP model validation report from 2018 https://emep.int/publ/reports/2018/sup_Status_Report_1_2018.pdf comparing EMEP model results to measurements for 2016. We have included a reference to this publication, and some accompanying text, just before section 3.1 (Se also reply to reviewer 1.)

11. Page 6, line 33: This discussion would benefit from a quantitative indication than just describing the largest contribution are seen for smaller countries with long coastline. I suggest to add a quantitative assessment like the contribution weighted by coastline length or weighted by distance-to-major-shipping-route to strengthen the statement here.

Answer: We agree that a quantitative assessment like the contribution weighted by coast-line length or weighted by distance-to-major-shipping-route would have strengthened the statement. We have considered this. The length of the coastline, and the coast/area ratios for countries are available from several sources such as Wikipedia and wold.by.map.org, all listing virtually identical numbers. We are however uncertain whether the methods calculating the length of the coastlines are comparable between countries. As an example all sources list the coastline of Estonia as being longer than both the coastlines of Sweden and Finland. This is most likely due to different measurement techniques. Basing our conclusions on data that are not comparable would not be scientifically sound. We have also considered using distance-to-major-shipping-route as a criteria, but found it hard to define and calculate in practice.

12. Page 7, line 6: Figure 3b shows the reduction of NO2 caused by ship emission in 2030 (i.e. For each country, the green bar along with blue bar is shorter than the green bar with black bar). As it is stated here, the improvement of the pollution levels is caused by reduction of BAS ship emissions. However, in page 3, line 24-29, you mentioned increase of ship size and fleets, and in page 4, line 23, the future scenario was assumed with NECA (and strict SECA?) applied. How the vessel size growth and fleet size increase, which would lead to more emissions, are balanced with strict regulations to have emission reduction in the future?

Answer: The future scenarios will either add or subtract vessel activity of the base year (2014), depending on the fleet size growth rate. If currently there were 100 containerships and an annual growth rate of one percent is applied, then 143 ship would exist in 2050. Adding 43 containerships to the fleet is done by replicating the activity of 43 randomly chosen containerships which exist in 2014. Introducing 43 new ships will need to comply with the existing year 2050 regulations, like Tier III limits, if the vessels were younger than 29 years. The changes in physical dimensions of future ships their impact on vessel speed/resistance curves is not considered, however.

13. Page 7, line 21: It was stated that increase of SOMO35 is more than annually

averaged ozone. The units of SOMO35 and annually averaged ozone are different. What is the comparing criterion in this statement?

Answer: We have added the word relative. This statement was based on considering percentage increases (which are not shown). Relatively speaking the changes in SOMO35 are more pronounced. This is not uncommon for a threshold indicator, where many areas are just below the threshold in the base case.

14. Page 7, line 23 & line 24: It was explained the changes in SOMO35 and annually averaged ozone by combination effect of ozone enhancement in the summer and decrease during the winter time. It would be supportive to add analysis of separating SOMO35 and ozone difference by two seasons into appendix and references to support the argument.

Answer: We have included figures of summer and winter SOMO35 and average ozone in the appendix and the discussion of the results for average ozone and SOMO35 is extended referring to these figures.

15. Page 7, line 25: There are some confusion for the discussion here. In Figure 4,both Germany and Denmark show decrease of annual mean ozone in 2030 (Present-2030, positive difference), but the statement here is "the additional emissions from BAS shipping lead to 'reductions' in annual ozone in Denmark. Furthermore,..... result in 'increased' annual ozone levels in Germany." Conflict arises from the differences between discussion mentioned above and Figure 4.

Answer: This part of the paper is re-written, See below

16. Page 7, line 27: It is not clear in the discussion here. Figure 4 shows that the-SOMO35 increases in the future (also stated in Page 7, line 21) for the two cases, but the statement here – "Even though annual ozone..... lower emissions will result in SOMO35 'reductions' in both these two cases.....". – you mentioned 'reduction' in SOMO35 instead. Additionally, I didn't see the clear connection between SOMO35

reduction and winter titration events.

Answer: We have now rewritten the text related to ozone and SOMO35. We have also included figures showing the winter and summer difference between the years 2030 and 2016 in ozone concentrations and SOMO35. We have have also added a reference corroborating our results.

Technical comments:

1. Page 7, line 9: It should be Figure 3e, instead of Figure 3d.2.

Answer: This is now fixed.

Page 7, line 19: Please rewrite the sentence, "Also show are the effects....".3. Figure 1: Please add X-axis and Y-axis label of longitude and latitude and remove the remaining cut-off headers in the plots.

Answer: Regarding line 19, page 7. This part is rewritten, see answer to previous comments.

X and Y long and Lat labels are added and cut-off headers are removed.
* * *

---

## Author Response (AR2)

Dear Editor

Below follows a point-by-point reply to the comments raised in the review (answers to the comments in italics).

**Comments to reviewer 1**

1. The abstract must provide quantitative information. This concerns both the country averaged contributions from ships and the change of pollutant concentrations between 2016 and 2030.

*We have added quantitative information for the contributions of Baltic Sea shipping to NO2 and PM2.5 levels in the abstract.*
*The last part of the abstract now reads:*
*"From Baltic Sea shipping the largest contributions are calculated for $NO_2$ in air, accounting for more than 50% in central parts of the Baltic Sea. In coastal zones contributions to $NO_2$ and also nitrogen depositions can be of the order of 20% in some regions. Smaller effects, up to 5 – 10%, are seen for $PM_{2.5}$ in coastal zones close to the main shipping lanes. Country averaged contributions from ships are small for large countries that extend far inland like Germany and Poland, and larger for smaller countries like Denmark and the Baltic states Estonia, Latvia and Lithuania, where ship emissions are among the largest contributors to concentrations and depositions of anthropogenic origin. Following the implementations of stricter SECA regulations, sulphur emissions from Baltic Sea shipping now have virtually no effects on $PM_{2.5}$ concentrations and sulphur depositions in the Baltic Sea region.*
*Adding to the expected reductions of air pollutants and depositions following the projected reductions in European emissions, we expect that the contributions from Baltic Sea shipping to $NO_2$ and $PM_{2.5}$ concentrations, and to depositions of nitrogen, will be reduced by 40 – 50% from 2016 to 2030 mainly as a result of the Baltic Sea being defined as a Nitrogen Emission Control Area from 2021. In most parts of the Baltic Sea region ozone levels are expected to decrease from 2016 to 2030. For the Baltic Sea shipping, titration, mainly in winter, and production, mainly in summer, partially compensate. As a result the effects of Baltic Sea shipping on ozone is similar in 2016 and 2030."*

2. The global emission dataset ECLIPSE version 5a was used, which has a horizontal resolution of 0.5 x 0.5 degrees. It needs to be specified how these coarse emissions have been resampled to the EMEP model grid of 0.1 x 0.1 degrees for Europe. Which proxies were used or was the emission data simply interpolated, ignoring smaller urban areas? It should also be stated how emissions of large point sources were handled.

*We have added more information on the re-gridding of the Eclipse emissions:*
*"The ECLIPSE v5a emissions were re-gridded using the TNO-MACC-III 0.125x0.0625*

*lon-lat emission distribution Kuenen et al. (2014) for year 2011. During the re-gridding process only the spatial distribution of the ECLIPSE v5a emissions was modified, while the national and sector totals remained unchanged. Where TNO-MACC-III emissions are not available (such as North-Africa) the gridded ECLIPSE v5a emissions were interpolated to the TNO-MACC grid resolution. Any missing sectors for countries which were included in the TNO-MACC-III emission data were also completed from the interpolated ECLIPSE v5a emissions."*

3. In the Conclusion (p.9, Line 24-27), a source-receptor calculation for secondary inorganic aerosol (SIA) is referred which was not presented in the paper. It needs to be clarified how the methodology of the present work (section 3.3) differs from the yearly reported source-receptor calculations in EMEP reports. Does the present study confirm the important contribution of the countries in the BAS region to levels of SIA (in PM2.5 and in PM10) in the countries themselves and in Europe?

*It should be noted that only the anthropogenic sources are considered in the source receptor calculations. We have included additional text here:*
*"EMEP source receptor calculations for the individual countries (see EMEP country reports for year 2016 (Klein et al. 2018)) show that, for many countries in the region, BAS shipping is among the 5 to 6 largest regions/countries contributing to SIA (Secondary Inorganic Aerosols). SIA is a major constituent of $PM_{2.5}$ typically ranging from about 30 to 60% of $PM_{2.5}$ mass in (scarce) measurements and in EMEP model calculations (Tsyro et al. 2018). Other constituents in $PM_{2.5}$ include seasalt and organics (both natural and anthropogenic) with no or minor contributions from shipping as well as primary particles. As a result, the percentage contributions from BAS shipping to SIA is of the order of a factor of two higher than for $PM_{2.5}$. As the natural part of $PM_{2.5}$ (and likewise $PM_{10}$) is not included in the EMEP source receptor calculations (EMEP 2018) they bears some resemblance to SIA. Thus the relative contributions from BAS shipping presented here is lower than the above source receptor calculations as it is compared to $PM_{2.5}$ (and likewise $PM_{10}$) of both antropogenic and natural origin. In a global model calculation with ship emission from the BAS and NOS also provided by FMI, source receptor relationships are in the same range as the reported EMEP results for 2014 and 2016 (Jonson et al. 2018). It should however be noted that the EMEP source receptor relationships are calculated by perturbing the emissions by 15%, whereas in this study we have excluded the emissions altogether in the NoShip scenarios."*

4. Figure 3c shows that only a very small fraction of PM2.5 is from ships. What is the contribution of ships to SIA in the countries of the BAS region and how high is it in the coastal zones?

*See discussion regarding point 3.*

**Comments to reviewer 3**

1. Page 6, line 30-34: The results showed that the measured and modeled fraction

of SO4 in PM2.5 only increase slightly with the Present_HiSulphur scenario. I suggest the authors provide possible explanations for the linkage between the SECA regulation and changes of SO4 and PM2.5 (Present_HiSulphur-Present_Base).

*We have included additional text here:*
*"Continuing a downward trend from the late 1980s, land-based sulphur emissions have decreased by more than 50%, i.e. more than for any other of the major air pollutants (Tista et al. 2018) and thus the importance of sulphur in particle formation has thus decresed relative to other anthropgenic emitted species and natural sources. In the SECAs the sulphur content in marine fuels has decreased from the global average of about 2.5% to 1% in 2011 and finally to 0.1% in 2015. As a result of these large emission reductions the fraction of $SO_4$ in $PM_{2.5}$ in the BAS region has decreased even further here. At the sites in Table 2 both the measured and model calculated fraction of $SO_4$ in $PM_{2.5}$ is about 0.15. As $SO_4$ make up a morerate portion of the $PM_{2.5}$ composition this fraction increase only by a small amount with the Present_HiSulphur scenario."*

2. Page 8, line 28-31: The statement In summer the increase caused by titration around the English channel..dominates the annual values. is confusing, as NOx titration leads to ozone reduction. Is the increase of SOMO35 and annual ozone over English Channel contrast to decrease of ozone for the rest BAS caused by reduction of titration? If it is what the authors mean, it would be helpful to clearly state the increase of ozone (SOMO35 and annual averaged ozone) and reduction of titration.

*We have added the word less. "... the increase caused by less titration around the English channel .."*

3. Page 8, line 31-34 & Page 9, line 1-2:
(1) It would be helpful to remind reader by referring the figure in detail for better understanding the discussion. For example, for the paragraph As shown in Figure 4.are higher in 2030 (but SOMO35 is reduced), you can add (green bar) to specifically refer the bar in Figure 4.

*We have added (green bars) when referring to Figure 4 here.*

(2) The statement In 2030 the additional emissions from BAS shipping..except Denmark does not match what Figure 4 shows. This sentence seems to discuss the blue bar (contributions from BAS in 2030) in Figure 4 but all the countries have increased SOMO35 and annually average ozone due to BAS ship emissions in the future, including Denmark. Also, the following sentence Here average ozone decreases (in contrast to the case in 2016, where SOMO35 increases when adding the emissions from BAS shipping) is not clear either.

*The reviewer is right in pointing out an error here. We have deleted the above mentioned sentence. This part now reads:*
*"In 2030 the additional emissions from BAS shipping result in increased SOMO35 and annually average ozone in all countries. (blue bars in Figure 4)."*

4. Page 10, line 19-25: The authors discussed whether the target set by IMO could be achieved in the future based on the results of this paper. Here the connections between the IMO target and the results of this paper are not clear, given the IMO target described here mainly focus on CO2 emissions and GHG emissions from shipping, while this paper focused on NOx, SO2, and PM2.5. The linkages between IMO target and the findings in this study can be stated more clearly.

*We have tried to make the meaning more clear.*
*"It is unlikely that this goal can be reached without substantial penetration of zero emission ships. If a portion of these zero emission ships run on electricity or hydrogen in 2030 they will be zero emission also for sulphur, nitrogen and $PM_{2.5}$ (in addition to $CO_2$), potentially resulting in reductions of these air pollutants beyond what is assumed in the Future_Base scenario in this paper."*

**Technical comments**

1. Page 3, line 1: Please specify what LNG stands for.
*Added in brackets (Liquefied Natural Gas)*
2. Page 17, Table2: The Base-model calculated values of NO2 and SO2 for the site Uto are negative.
*Corrected. (Calc. and NMB had swapped places)*
3. Page 18, Table2: The HiSulphur-model calculated value of SO4 for the site Rao is negative.
*Corrected (Calc. and NMB had swapped places)*
3. Page 23, Figure 4: The figure descriptions and legends are not consistent (green bar should be Present_Base  Future_Base; blue bar should be Future_Base  Future_NoShip).
*Corrected. We changed the colours in the plot to enhance readability of the text in figure a. Unfortunately we did not change the legend.*

---

## Author Response (AR3)

Dear Editor

Below follows our final corrections following the comments raised in the review (answers to the comments in italics).

Regarding the presentation quality it is unclear if anything or what is required here.

Non-public comments to the Author:

Both referees require technicdal corrections. In addition, one referee requires to improve the presentation quality. Please see the comments attached and address the concerns carefully.

**Comments to reviewer 1**

I have no further comments. The manuscript can be published.

Technical Correction:

Figure 3. In the last sentence of the caption: numeric values of NO2 Add Baltic 2014 and SO2 Add Baltic are not shown.

*This correction has been made:*

*"Numeric values for* $NO_2$ *Add Baltic 2014 and for* $SO_2$ *Add Baltic not given as they are very small"*

**Comments to reviewer 3**

The authors have addressed the reviewers comments and revised the manuscript accordingly. This paper clearly demonstrates the effect of Baltic Sea ship emissions on air quality and deposition under the SECA regulations and future NECAs, which has important implications for future emission regulations. The revised abstract and some paragraphs are helpful to better understand the results and to highlight the importance of this study. However, after this revision, I found there is one point that is necessary to be clarified before publication:

1. Page 9, line 11-14: In the manuscript, it states As shown in Figure 4 (green bars) from year 2016 to 2030 result in overall reductions in ozone levels.. Based on this statement, my understanding is that lower ozone level is expected in the future (that is: Future_Base Present_Base ¡ 0) due to emission reductions in the future. The green bars in figure 4 also show negative ozone difference. However, the label and the caption of green bar are Present_Base Future_Base, which is conflict to what it describes in the manuscript. To avoid confusion, the authors should check the label/caption or clarify the statement in the manuscript.

*We have corrected the caption in Figure 4 so that :*

*"..... ozone green bars represent changes in levels from 2016 to 2030 (Future_Base – Present_Base), ........"*